# Blocking interaction between SHP2 and PD-1 denotes a novel opportunity for developing PD-1 inhibitors

Zhenzhen Fan[1,†], Yahui Tian[1,†], Zhipeng Chen[1], Lu Liu[1], Qian Zhou[1], Jingjing He[2], James Coleman[3] (iD), Changjiang Dong[3] (iD), Nan Li[1], Junqi Huang[1], Chenqi Xu[4], Zhimin Zhang[5], Song Gao[2] (iD), Penghui Zhou[2,*] (iD), Ke Ding[5,**] (iD) & Liang Chen[1,6,***] (iD)

## Abstract

Small molecular PD-1 inhibitors are lacking in current immuno-oncology clinic. PD-1/PD-L1 antibody inhibitors currently approved for clinical usage block interaction between PD-L1 and PD-1 to enhance cytotoxicity of CD8[+] cytotoxic T lymphocyte (CTL). Whether other steps along the PD-1 signaling pathway can be targeted remains to be determined. Here, we report that methylene blue (MB), an FDA-approved chemical for treating methemoglobinemia, potently inhibits PD-1 signaling. MB enhances the cytotoxicity, activation, cell proliferation, and cytokine-secreting activity of CTL inhibited by PD-1. Mechanistically, MB blocks interaction between Y248-phosphorylated immunoreceptor tyrosine-based switch motif (ITSM) of human PD-1 and SHP2. MB enables activated CTL to shrink PD-L1 expressing tumor allografts and autochthonous lung cancers in a transgenic mouse model. MB also effectively counteracts the PD-1 signaling on human T cells isolated from peripheral blood of healthy donors. Thus, we identify an FDA-approved chemical capable of potently inhibiting the function of PD-1. Equally important, our work sheds light on a novel strategy to develop inhibitors targeting PD-1 signaling axis.

**Keywords** immunotherapy; methylene blue; PD-1; small molecular inhibitor; transgenic mouse model
**Subject Categories** Cancer; Immunology

## Introduction

Antibodies against PD-1 or PD-L1 prevent interaction between PD-1 and its ligand, block the activation of PD-1, and thereby restore the activity of CTL to lyse PD-L1 expressing target cells and shrink tumor *in vivo* (Hirano *et al*, 2005). Some of these antibodies have been approved for clinical trial or usage (Dolan & Gupta, 2014). Impressive clinical benefit was seen in a portion of cancer patients (Brahmer *et al*, 2010; Topalian *et al*, 2012; Powles *et al*, 2014).

Unfortunately, PD-1 inhibitors currently used in clinic show severe, sometimes fatal, side effects (Johnson *et al*, 2016; Moslehi *et al*, 2018). In contrast, PD-1-deficient mice of C57BL/6 background develop and grow apparently normally, with autoimmune disease observed in late stage of their life (Nishimura *et al*, 1998, 1999). Lines of evidence suggest that the fragment crystallizable regions (Fc region) of these antibodies play a role in mediating treatment effect (Dahan *et al*, 2015). These data suggest that it is antibodies, rather than PD-1 inhibition per se, that cause the side effects currently seen in clinic. Small molecular PD-1 inhibitors are, therefore, expected to achieve therapeutic effect comparable to antibody drugs while eliminating the toxicity issues. Unfortunately, small molecular PD-1 inhibitors are lacking in the current immuno-oncology clinic. Moreover, significant portion of the current PD-1 drugs inhibit PD-1 function through blocking interaction between PD-1 and PD-L1. Whether other steps along the PD-1 signaling pathway can be targeted to enhance cytotoxicity of CD8[+] T cells remains to be determined.

1 MOE Key Laboratory of Tumor Molecular Biology and Key Laboratory of Functional Protein Research of Guangdong Higher Education Institutes, Institute of Life and Health Engineering, Jinan University, Guangzhou, China
2 Sun Yat-Sen University Cancer Center, Guangzhou, China
3 Biomedical Research Centre, Norwich Medical School, University of East Anglia, Norwich Research Park, Norwich, UK
4 State Key Laboratory of Molecular Biology, Shanghai Science Research Center, CAS Center for Excellence in Molecular Cell Science, Shanghai Institute of Biochemistry and Cell Biology, Chinese Academy of Sciences, University of Chinese Academy of Sciences, Shanghai, China
5 School of Pharmacy, Jinan University, Guangzhou, China
6 The First Affiliated Hospital of Jinan University, Guangzhou, China
  *Corresponding author. Tel: +86 20 20 8734 3392; Fax: +86 20 20 8734 3392; E-mail: zhouph@sysucc.org.cn
  **Corresponding author. Tel: +86 020 8522 0850; Fax: +86 020 8522 4766; E-mail: dingke@jnu.edu.cn
  ***Corresponding author. Tel: +86 20 20 8522 2875; Fax: +86 20 20 8522 7039; E-mail: chenliang@jnu.edu.cn
  †These authors contributed equally to this work

Signaling pathways leading from binding of PD-L1/2 to PD-1 down to inhibition of T-cell activation and cytokine expression have been relatively clear (Riley, 2009; Xia et al, 2016). Although SHP1 and SHP2 were earlier reported to be recruited by PD-1 to exert immunoinhibitory function, it is been confirmed that SHP2 is the main effecter (Yokosuka et al, 2012). Upon stimulation by PD-L1, immunoreceptor tyrosine-based switch motif (ITSM) in the cytoplasmic tail of human PD-1 is phosphorylated on Y248, whereby providing a docking site for Src Homology 2 (SH2) domain of SHP2, which is essential for interaction between PD-1 and SHP2 (Yokosuka et al, 2012). Reports have shown that phosphorylation of CD28 in T cells is critical for mediating treatment effect of PD-1 inhibition (Hui et al, 2017; Kamphorst et al, 2017).

Protein–protein interactions (PPI) were previously considered "undruggable" by small molecular chemicals. Recent mutational study, however, revealed "hot spots" anchor residues that contributed the most to the binding free energy of the protein–protein complex (Jin et al, 2014). By placing molecules at these sites, orthosteric (i.e., competitive) small molecular inhibitors can effectively block interaction of the targeted two proteins. Currently, PPIs are targeted by an increasingly larger number of small molecular chemicals (Ran & Gestwicki, 2018). Impressive successes have been reported with inhibitors of protein–protein interactions (iPPIs) in clinic as exemplified by Bcl-2 inhibitor, Venclexta (ABT-199) (Mihalyova et al, 2018).

Methylene blue (MB) is an FDA-approved small molecular drug, used to treat patients with methemoglobin levels greater than 30% or those who have symptoms despite oxygen therapy (Committee, 2015). It has previously been used for cyanide poisoning and urinary tract infections, which is no longer recommended. MB is known for its highly favorable safety profile as reflected in a recent study showing that MB can be safely administered to reach a serum concentration up to 6 μM (Baddeley et al, 2015).

In our current study, we report that MB effectively counteracted the suppressive activity of PD-1 on CTLs and restored their cytotoxicity, activation, proliferation, and cytokine-secreting activity. Mechanistically, MB blocked interaction between SHP2 and Y248-phosphorylated ITSM motif of human PD-1 and thus potently inhibited the recruitment of SHP2 by PD-1 in CTLs when stimulated by PD-L1. Impressive antitumor effect of MB was seen in allograft and genetically engineered mouse tumor models. MB also recovered proliferation and cytokine expression by human CD8[+] T cells. Our work therefore not only identified a potent small molecular iPPI for blocking interaction between PD-1 and SHP2, but shed light on a novel strategy for developing inhibitors targeting PD-L1/PD-1 signaling axis.

# Results

## MB enhances cytotoxicity of activated CTL against PD-L1 expressing target cells

We took advantage of engineered T-cell systems in which luciferase gene was placed under control of NFAT binding sequence, such that luciferase activity could serve as surrogate IL-2 mRNA transcription (Chow et al, 1999). To identify a chemical PD-1 inhibitor through a high-throughput screening, we generated a system composed of a stable Jurkat T-cell line expressing human PD-1 and harboring NFAT-luciferase reporter (designated JP-luc for Jurkat-PD-1-NFAT-luciferase) and a stable Raji cell line (an antigen-presenting cell) expressing human PD-L1 (designated Raji-L1) (Fig EV1A, Appendix Fig S1A and B). When co-incubated with superantigen SEE-loaded parental Raji, JP-luc exhibited robust luciferase activity. In contrast, JP-luc showed limited luciferase activity when co-cultured with SEE-loaded Raji-L1 (Appendix Fig S1C). Treatment with PD-1 antibody (designated aPD1) recovered the luciferase activity in JP-luc in the presence of SEE-loaded Raji-L1. We thus established an efficient system to assay the intensity of PD-1 signaling by monitoring luciferase activity of JP-luc (Appendix Fig S1C) and whereby screened our chemical library. We identified 3,7-bis(dimethylamino)-5-phenothiazinium chloride (also called methylene blue; designated MB) as the best hit (Figs 1A and EV1B).

We then assayed MB's ability to enhance cytotoxicity of PD-1 expressing CTL using in vitro cellular system. E.G7-OVA (designated EG-7) is a cell line derived from spontaneous mouse thymoma cell, EL-4, through stably transfecting with the complementary DNA of chicken ovalbumin (OVA). This cell line presents OVA with an H-2K[b]-restricted CTL epitope (SIINFEKL) that is recognized by OT-1 transgenic TCR (Moore et al, 1988). We first generated an EG-7 cell clone stably expressing PD-L1 (referred hereafter to as EG7-L1) (Appendix Fig S1D). SIINFEKL peptide stimulation rapidly activated splenic cells of OT-1 mice and expanded CD8[+] T cells population to a purity of almost 100% within 3 days (Appendix Fig S1E). OT-1

**Figure 1. MB enhances cytotoxicity of activated CTL against PD-L1 expressing target cells.**

A   Structural formula of 3,7-bis(dimethylamino)-5-phenothiazinium chloride (MB).

B   Impact of MB on cytotoxicity of OT-1 CTLs against EG7 or EG7-L1. Splenocytes from OT-1 mice in culture were stimulated with 10 nM of OVA$_{257-264}$ (SIINFEKL) for 3 days to generate mature CTLs. CTLs were incubated with CFSE-labeled EG7-L1 cells in the presence of MB at indicated concentrations. Cytotoxicity was determined by flow cytometry analysis. Data are representative of three independent experiments (effector-to-target ratio = 5:1, unpaired t-test). EG7-L1: EG7 overexpressing PD-L1.

C   Statistical results of (B).

D   ELISA measured of LDH release to assess the cytotoxicity of OT-I CTLs on EG7-L1 in the presence of MB. Anti-PD1 antibody (aPD-1) served as positive control.

E   Impact of MB on cytotoxicity of PD-1$^{-/-}$ (PD-1KO) OT-1 CTLs against EG7 or PD-L1 expressing EG7 stable cell line. Data are representative of three independent experiments (unpaired t-test).

F   Statistical results of (E).

G   Cytotoxicity of OT-I CTLs against B16-F10-OVA cells (designated B16-OVA) in the presence of 1 μM of MB or DMSO. Cytotoxicity was determined by the relative area unoccupied by cells examined under microscope (scale bar = 50 μm). Data are representative of three independent experiments and were analyzed by unpaired t-test.

H   Statistical results of (G).

Data information: Data are representative of three independent experiments. Unpaired t-test; error bars denote SEM. **P < 0.01; ***P < 0.001; ****P < 0.0001.

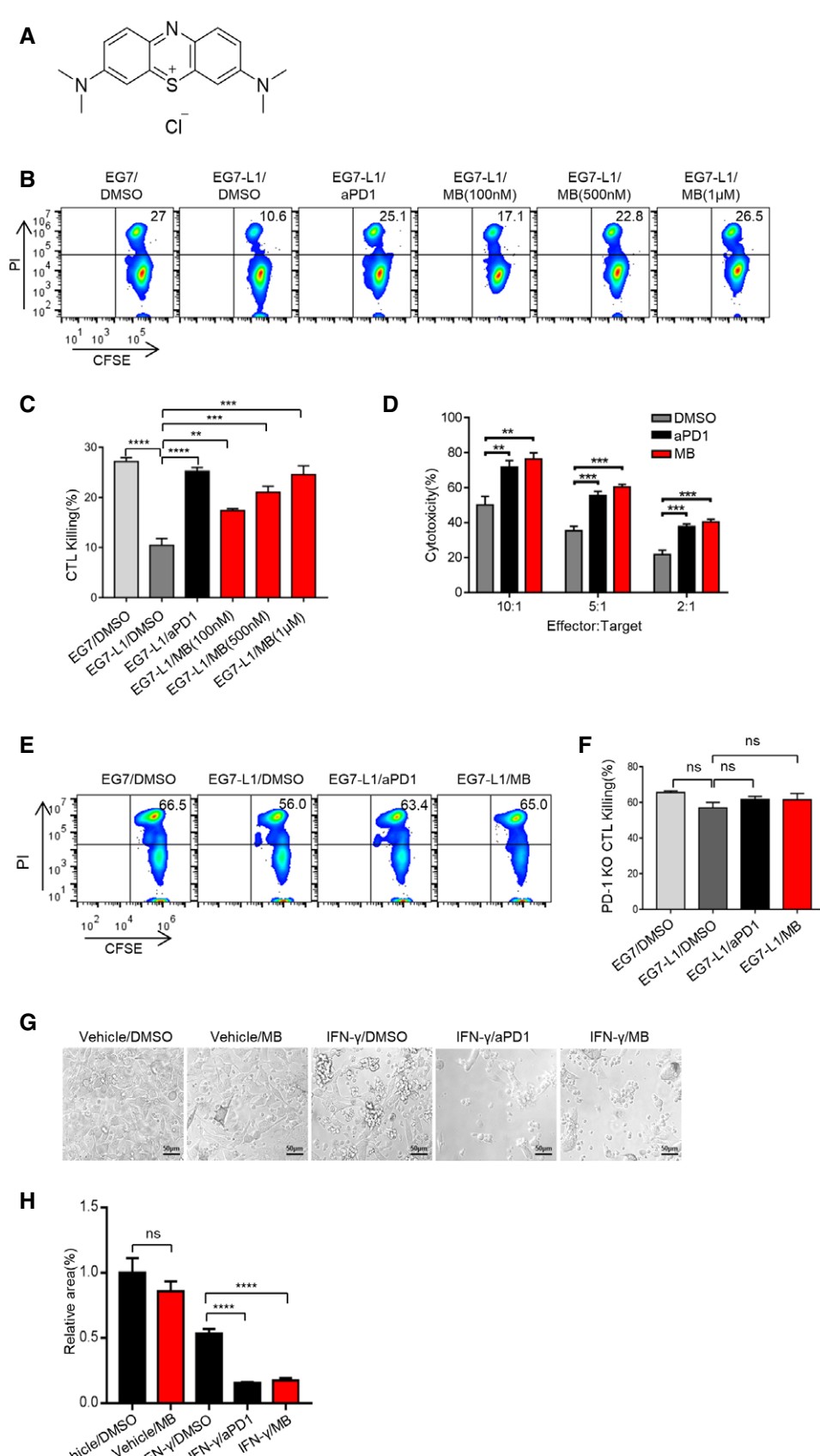

**Figure 1.**

CTL rapidly upregulated PD-1 expression during this activation (Appendix Fig S1F). We found that OT-1 CTL exhibited limited cytotoxicity against EG7-L1 target cells, which was significantly enhanced by administration of aPD1 (Appendix Fig S1G, Fig EV1C and D). FACS analysis showed that MB potently recovered cytotoxicity of CTL in a dose-dependent manner (Fig 1B and C). Moreover, this cytotoxicity was repeated at various effector-to-target ratios (Appendix Fig S1H and I). Of note, concentrations of MB in these experiments are well below clinically achievable serum concentration. Moreover, we saw no cytotoxicity of MB at these concentrations toward EG-7 cells (Appendix Fig S1J). Measuring lactate dehydrogenase (LDH) released by target tumor cells into medium is another accurate assay to determine CTL cytotoxicity (Decker & Lohmann-Matthes, 1988). We quantified the LDH level in media and confirmed the effectiveness of MB in enhancing cytotoxicity of PD-1-positive OT-1 CTL against EG7-L1 (Fig 1D). To unequivocally show that PD-1 was a critical target mediating the cytotoxicity enhancing effect of MB, we repeated this experiment with OT-1 CTLs of PD-1$^{-/-}$ (PD-1KO) background (Nishimura *et al*, 1998) (Appendix Fig S1K and L). Critically, we found that MB treatment did not further significantly enhance cytotoxicity of PD-1KO OT-1 CTLs against EG7-L1 (Fig 1E and F).

We went on to validate the effectiveness of MB to enhance cytotoxicity of OT-1 CTL against mouse melanoma cell B16-F10 (designated B16 cells). We confirmed the ability of IFNγ to induce PD-L1 expression in B16 cells (Appendix Fig S1M). Earlier reports also revealed that surface expression of MHC class I by B16 cells was strikingly upregulated by IFNγ (Bohm *et al*, 1998). Hence, OT-1 CTL could only recognize and effectively kill IFNγ-treated B16-OVA, but not B16 cell or B16-OVA untreated with IFNγ. Consistently, OT-1 CTL exhibited no obvious cytotoxicity against B16 or B16-OVA and limited cytotoxicity against IFNγ-treated B16-OVA while aPD1 treatment significantly enhanced OT-1 CTL's cytotoxicity against IFNγ-treated B16-OVA (Appendix Fig S1N and O). Using this system, we found that administration of MB potently enhanced cytotoxicity of OT-1 CTL against IFNγ treated B16-OVA (Figs 1G and H, and

EV1E). Taken together, MB enhanced cytotoxicity of PD-1-positive CTL against PD-L1 expressing target cells.

## MB enhances proliferation and activation of CTL

Proliferation and activation are critical for function of CTLs. We asked whether MB enhances both aspects of CTLs. We first checked the ability of MB to promote proliferation of T cells in the presence of PD-L1. CFSE-labeled splenic cells were stimulated with aCD3/aCD28 and proliferation, as indicated by dilution of CFSE, of T cells of wild type (WT), and PD-1KO background was analyzed 48 h after stimulation through FACS analysis by gating on CD8$^+$ population. Data showed that proliferation of WT T cells was significantly suppressed by PD-L1 treatment and this suppression was eliminated by administration of aPD1 or MB (Fig EV2A and B, Appendix Fig S2A). In contrast, neither MB nor aPD1 significantly enhanced proliferation of PD-1KO CTLs, arguing that MB, just like aPD1, enhanced proliferation of T cell through inhibiting PD-1 (Fig 2A and B).

We further checked the ability of MB to activate T cells. Expression of CD25 (Theze *et al*, 1996) and CD69 (Santis *et al*, 1992) is 2 typical activation markers for T lymphocytes. We found that splenic cells started to express CD25 when cultured in aCD3/aCD28-coated plates, which was effectively suppressed by PD-L1 treatment. aPD1 administration eliminated this suppression. FACS analysis revealed that 100 nM of MB potently recovered CD25 expression in CTLs suppressed by PD-L1 in this system (Fig 2C and D, Appendix Fig S2B). Similar effect was seen on CD69 expression (Fig 2C and D). These data showed that MB could restore the level of activation of CTLs suppressed by PD-1.

## MB enhances effector functions of CTL

T cells start to secrete IL-2 in response to TCR signaling. We used JP-luc and Raji-L1 stable cell lines to quantitatively measure the suppressive role of PD-1 signaling on IL-2 expression in T cells by

**Figure 2. MB enhanced activation and effector function of CTL.**

A   Effect of MB on the proliferation of WT CTLs and PD-1KO CTLs. Splenic cells were stained with 5 μM of CFSE and seeded into aCD3/aCD28-coated 96-well plates. 10 μg/ml of PD-L1 was administered in media. Cell proliferation was checked by monitoring dilution of CFSE 48 h after CD3/CD28 stimulation by gating on CD8-positive population through FACS analysis. WT: splenic cell from wild-type C57BL/J mice. PD-1KO: splenic cell from PD-1 knockout mice.

B   Statistical results of (A).

C   FACS analysis of the effect of MB on the activation of OT-I CTLs. OT-I CTLs were stimulated with precoated aCD3/aCD28 and 10 μg/ml of mouse PD-L1 protein in the presence of 100 nM MB or 10 μg/ml aPD1 (served as positive control). After 24 h, surface expression of CD25 and CD69 on OT-1 CTLs was determined through FACS analysis.

D   Statistical results of (C).

E   Effect of MB on the activation status of Jurkat T cells. Luciferase activity is suppressed in JP-luc by co-culture with Raji-L1 preloaded with 1 μg/ml superantigen (SEE). Treatment with MB enhanced luciferase expression (SEE-loaded Raji (PD-L1 negative) served as positive control). IC$_{50}$ is calculated to be 117 nM. JP-luc: Jurkat cell harboring NFAT-luciferase transgene and overexpressing PD-1; Raji-L1: Raji overexpressing PD-L1.

F   Impact of MB on luciferase activity of various engineered Jurkat T cells. J-luc: Jurkat cell harboring NFAT-luciferase transgene; J-luc-sgPD-1: J-luc cells treated with lentivirus expressing sgPD-1/CAS9 simultaneously.

G   qRT–PCR analysis of IL-2 mRNA level in JP-luc cells stimulated with precoated aCD3/aCD28 (10 μg/ml) in the presence of 10 μg/ml of PD-L1 and MB at indicated concentrations.

H   MB enhancing IL-2 expression by JP-luc stimulated with Raji-L1 for 24 h quantified through ELISA analysis (aPD1 served as positive control).

I   MB enhancing production of cytokine and cytolytic granule by OT-I CTLs. CTLs were incubated with EG7-L1 cells in the presence of protein transport inhibitor (PTI) and MB at indicated concentrations. Expression of cytokine and cytolytic granule was determined by flow cytometry. aPD1 served as positive control. EG7-L1: EG7 overexpressing PD-L1.

J   Statistical results of (I).

Data information: Data are representative of three independent experiments. Unpaired *t*-test; error bars denote SEM. *P < 0.05; **P < 0.01; ***P < 0.001; ****P < 0.0001.

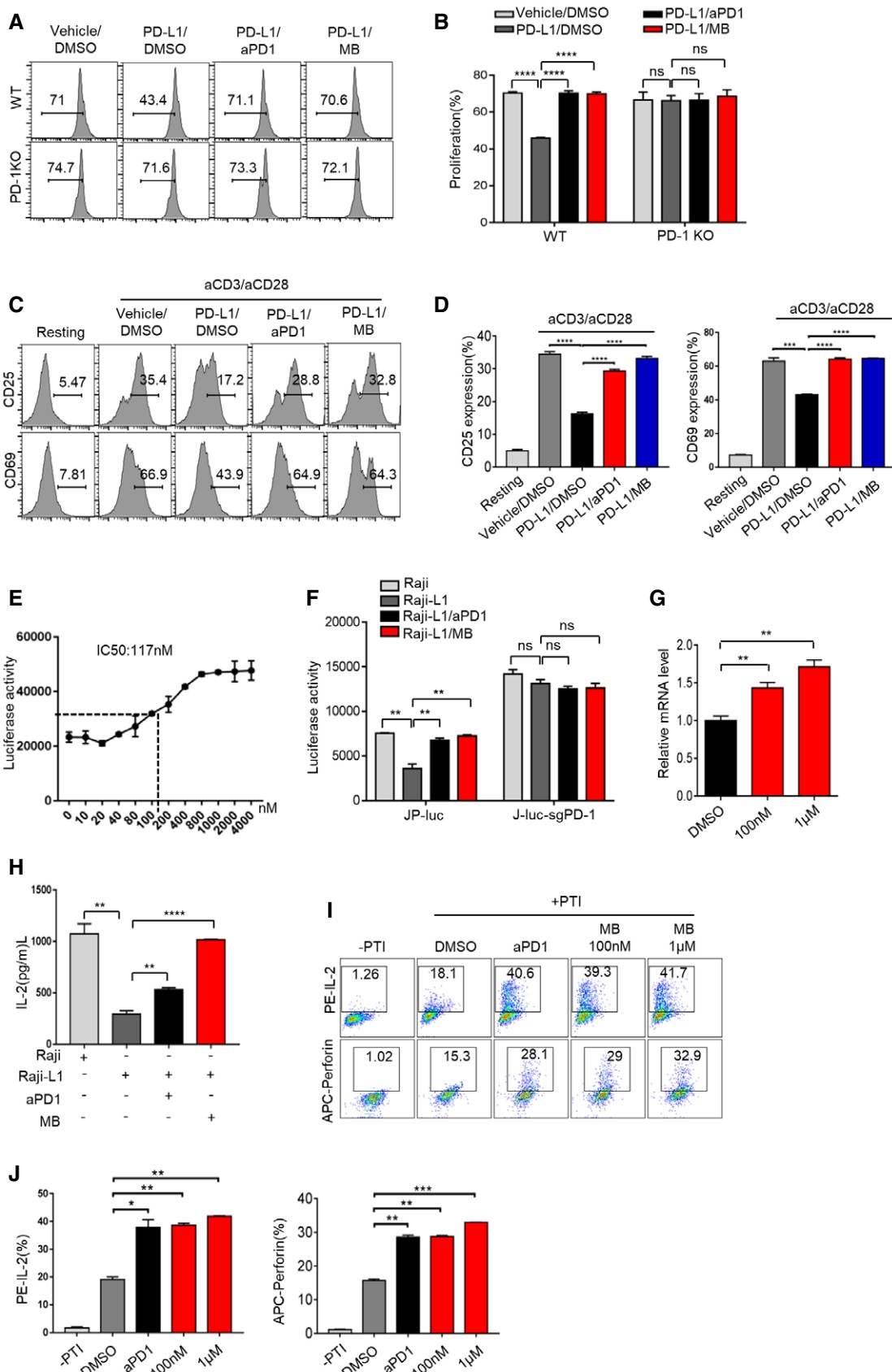

**Figure 2.**

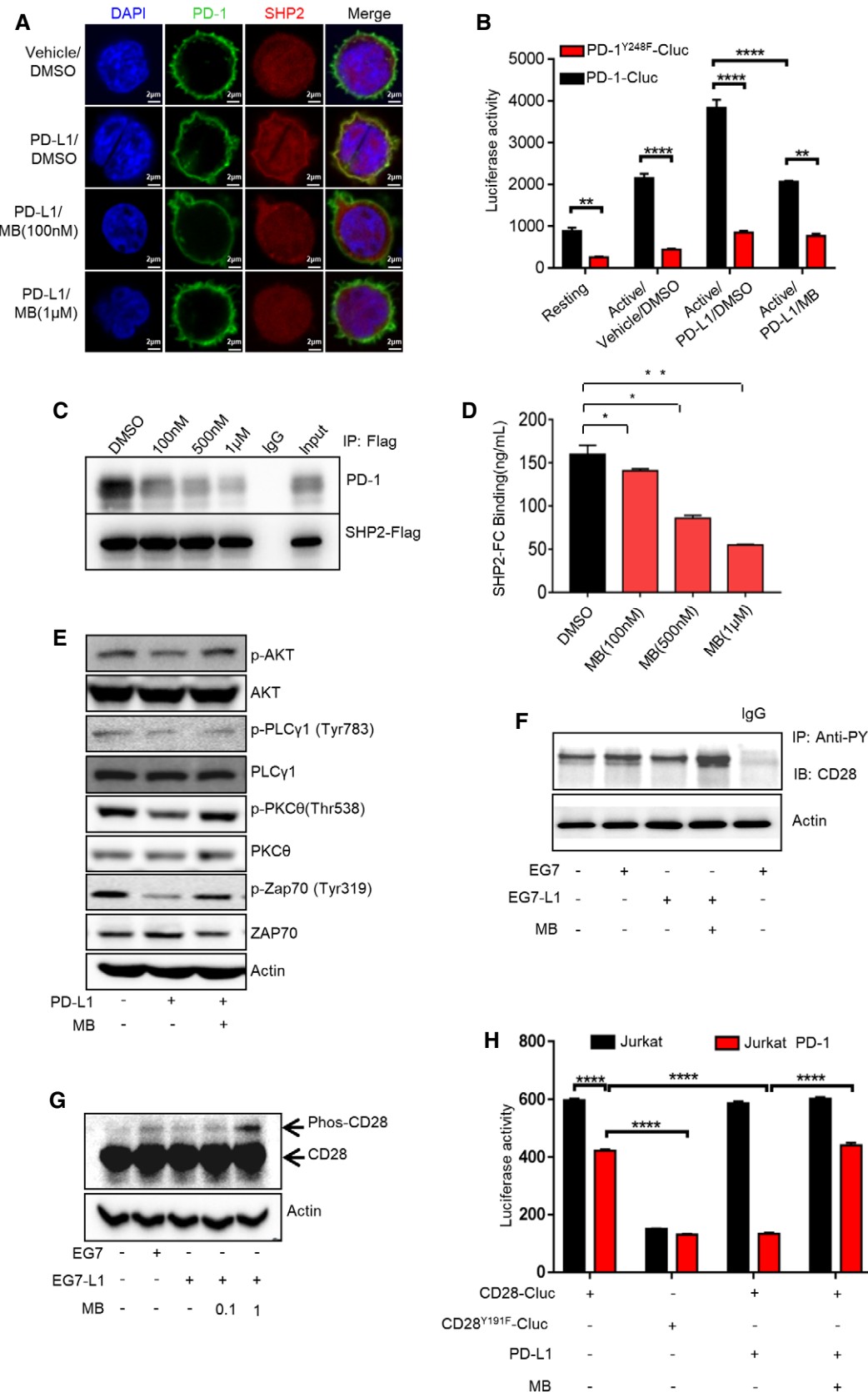

**Figure 3.**

**Figure 3.  MB suppress PD-1 signaling through blocking SHP2 recruitment by PD-1.**

A   Confocal cell images of Jurkat cells co-expressing PD-1-EGFP and SHP2-mCherry. Jurkat cells co-expressing PD-1-EGFP and SHP2-mCherry were incubated with 10 μg/ml of human PD-L1 protein in the presence of 100 nM and 1 μM of MB for 2 min, fixed with 4% paraformaldehyde, and stained with DAPI. Colocalization signal was examined with confocal microscope (scale bar = 2 μm).

B   Fluorescence complementation analysis of the effect of MB on the interaction between PD-1 and SHP2. Jurkat cells co-expressing PD1-C-Luc or PD1$^{Y248F}$-C-Luc and SHP2-N-luc were seeded in a 96-well plate precoated with 10 μg/ml of aCD3/aCD28 with media supplemented with 10 μg/ml human PD-L1 protein in the presence of MB at indicated concentrations for 6 h followed by analysis of luciferase activity.

C   Co-IP analysis of impact of MB on interaction between PD-1 and SHP2. 293T cells co-expressing PD-1 and SHP2-FLAG were treated with MB at indicated concentration for 6 h.

D   Ability of MB to block interaction between PD-1 with SHP2 revealed through ELISA. Y248-phosphorylated human PD-1 ITSM peptide was coated at 4 μg/ml in 96-well plate overnight. Recombinant human SHP2 protein incubated with the coated peptide in the presence of MB for 2 h. Bound SHP2 was measured by monitoring the activity of horse peroxidase conjugated on secondary antibody targeting SHP2.

E   Representative Western blot showing the levels of total and phosphorylated ZAP70, PKCθ, and PLCγ1 in the lysates of stimulated CTL, inhibited by PD-L1 in the presence of 1 μM of MB.

F   Representative Western blot showing the levels CD28 phosphorylation. CTL was stimulated with EG7 cells (parental), EG7-L1, and EG7-L1 in the presence of 100 nM of MB for 2 h. Protein immunoprecipitated (IP) with anti-pY from the lysates of the indicated CTL-EG7 co-culture was subjected to SDS–PAGE separation and blotted with aCD28. β-Actin served as loading control. EG7-L1: EG7 overexpressing PD-L1.

G   Phos-tag analysis of CD28 phosphorylation in CTL. CTLs were stimulated with EG7 or EG7-L1 in the presence of 1 μM MB. Cell lysate was then separated by SDS–PAGE containing 50 μM of Phos-tag and blotted with aCD28 antibody. Phosphorylated CD28 (slow-moving) species were visualized through exposure.

H   Fluorescence complementation analysis of the effect of MB on the interaction between CD28 with P85. Jurkat or Jurkat-PD-1 cells were co-transfected with CD28-C-Luc or CD28$^{Y191F}$-C-Luc and P85-N-Luc. Cells were seeded in a 96-well plate coated with 10 μg/ml of aCD3/aCD28 in the presence of human 10 μg/ml of PD-L1 protein and 100 nM of MB for 6 h. Luciferase activity was measured as readout for CD28-P85 interaction.

Data information: Data are representative of three independent experiments and were analyzed by unpaired *t*-test. Error bars denote SEM. *$P < 0.05$; **$P < 0.01$; ****$P < 0.0001$.
Source data are available online for this figure.

monitoring luciferase activity. In this system, MB potently counteracted the suppression imposed by PD-L1 on JP-luc with an IC$_{50}$ of 117 nM (Fig 2E). Critically, in a parallel experiment with Jurkat cells harboring NFAT-luciferase and CRISPR/CAS9-mediated knockout of PD-1 (designated J-luc-sgPD-1), we found MB did not further enhance luciferase activity (Fig 2F, Appendix Fig S2C). The transcription of IL-2 mRNA was verified through qRT–PCR (Fig 2G). We also confirmed secretion of IL-2 by measuring protein concentration in the media through ELISA and found the same pattern (Fig 2H).

We went on to check the ability of MB to enhance the expression of key effector proteins, IL-2, perforin, IFNγ, and granzyme B by OT-1 CTL in the presence of PD-L1. As expected, aPD1 significantly enhanced expression of effector molecules by CTLs stimulated with EG7-L1 (Figs 2I and J, and EV2C–E). In a parallel experiment, we found that MB dose-dependently enhanced protein level of IL-2 and perforin (Fig 2I and J) and granzyme B and IFNγ in T cells (Fig EV2C–E).

Together, our data convincingly showed that MB restored effector function of CTLs suppressed by PD-1.

### MB inhibits the function of PD-1 through a novel mechanism

A significant portion of current PD-1 inhibitors function through blocking interaction between PD-1 and its ligand. We went on test whether MB blocked interaction between PD-1 and PD-L1. To this end, we assayed the impact of MB on the binding of PD-L1-Fc fusion protein to Jurkat-PD-1 cells through FACS analysis. However, we found that MB did not interfere the binding of PD-L1 to PD-1 expressing cells (Fig EV3A, Appendix Fig S3A). Using purified recombinant extracellular domain of PD-1 and PD-L1-Fc proteins, we conducted an ELISA assay to quantify the impact of MB on binding of PD-L1 to PD-1 coated on plates. Results showed that MB did not interfere interaction between both human and mouse PD-1 and their respective ligands (Appendix Fig S3B). Thus, MB inhibited

function of PD-1 through a novel mechanism other than blocking interaction between PD-1 and PD-L1.

### MB suppresses PD-1 signaling through blocking recruitment of SHP2 by PD-1

We then checked the inhibition of signaling events further downstream along the PD-1 pathway. PD-1 recruits SHP2 to inhibit the TCR signaling when stimulated by PD-L1 (Riley, 2009; Yokosuka *et al*, 2012; Xia *et al*, 2016). To find out whether MB interferes interaction between PD-1 and SHP2, we transfected Jurkat cells with EGFP-tagged PD-1 and mCherry-tagged SHP2. Confocal microphotography analysis confirmed cell membrane localization for EGFP and cytoplasmic localization for mCherry, corresponding to cellular localization of PD-1 and SHP2 in T cells (Fig 3A, Appendix Fig S3C–G). Earlier reports showed that PD-L1 stimulation led to phosphorylation of Y248 in ITSM motif of PD-1, essential for recruitment of SHP2 to inhibit TCR signaling (Yokosuka *et al*, 2012). Consistently, we found that treatment of above transfected Jurkat cells with PD-L1 resulted in Y248 phosphorylation of PD-1 (Appendix Fig S3H). Moreover, PD-L1 treatment enhanced colocalization of mCherry to membrane EGFP, indicating recruitment of SHP2 by PD-1 (Fig 3A, Appendix Fig S3C–G). These results confirmed that our Jurkat system recapitulated the physiological process of PD-1 signaling in T cells. Using this assay system, we found that MB dose-dependently blocked the colocalization of EGFP/mCherry induced by PD-L1 treatment (Fig 3A, Appendix Fig S3C and D). We also found that treatment with pervanadate (PVD), a potent nonselective phosphatase inhibitor, resulted in Y248 phosphorylation of PD-1 and enhanced recruitment of SHP2 by PD-1 in Jurkat-PD-1-EGFP/mCherry-SHP2 cells (Appendix Fig S3E and I). Importantly, MB dose-dependently blocked the colocalization of EGFP/mCherry induced by PVD treatment (Appendix Fig S3E–G).

To quantitatively measure the ability of MB to block interaction between PD-1 and SHP2, we turned to bimolecular fluorescence

complementation assay (Shyu & Hu, 2008) by fusing PD-1 and SHP2 to C- and N-terminal half of firefly luciferase respectively (designated PD-1-C-luc and SHP2-N-luc, respectively). We then co-expressed PD-1-C-Luc and SHP2-N-luc in Jurkat cells, such that luciferase activity was a direct readout for interaction between PD-1 and SHP2 in Jurkat cells. Result showed that MB potently inhibited interaction of PD-1 with SHP2 elicited by PD-L1 treatment (Fig 3B).

To further confirm the ability of MB to block interaction of SHP2 with Y248-phosphorylated PD-1 in an independent cell line through luciferase complementation assay, we generated stable 293T-cell clone co-expressing SHP2-N-luc and PD-1-C-luc (designated 293T-SHP2/PD-1). We detected baseline PD-1 Y248 phosphorylation in 293T-SHP2/PD-1, which was dramatically enhanced by PVD treatment (Appendix Fig S3J). PVD treatment enhanced luciferase activity in this 293T-SHP2/PD-1 cell line specifically depended on Y248 phosphorylation (Fig EV3B). We found that MB potently blocked interaction between SHP2 and Y248-phosphorylated PD-1 in this system (Appendix Fig S3K). We further sought to validate the ability of MB to block interaction between PD-1 and SHP2 through co-IP assay. To this end, we co-expressed SHP2-FLAG and PD-1 transiently in 293T cells and subjected the cell lysate to immunoprecipitation with FLAG antibody, followed by Western analysis with PD-1 antibody. We found that MB dose-dependently prevented interaction of PD-1 and SHP2 (Fig 3C, Appendix Fig S3L). These results solidly showed that MB functioned to prevent recruitment of SHP2 by PD-1.

To find out whether MB directly interfered the interaction between PD-1 and SHP2, we conducted ELISA assay with synthesized peptide covering Y248-phosphorylated ITSM of human PD-1 (designated Pep-ITSM-p248Y) and purified recombinant SHP2-Fc. We found that MB dose-dependently blocked the binding of SHP2 protein to coated Pep-ITSM-p248Y, unequivocally showing that MB directly blocked interaction between SHP2 and PD-1 (Fig 3D, Appendix Fig S3M).

### MB treatment enables effective TCR signaling in T cells

We then checked in detail the impact of MB on signaling events downstream of TCR in OT-1 CTL. We found that MB treatment restored phosphorylation of Akt, PLCγ, PKCθ, and ZAP70 in OT-1 CTL suppressed by PD-L1 treatment (Fig 3E).

CD28 phosphorylation in CTLs is a critical determinant of treatment effect of PD-1 inhibitors (Hui et al, 2017; Kamphorst et al, 2017). We stimulated OT-1 T cells with EG-7 or EG7-L1, and total cell lysate was subjected to pulldown with anti-phosphotyrosine antibody followed by immunoblot with CD28 antibody to check phosphorylation of CD28. Our data showed that MB strongly enhanced phosphorylation of CD28 in OT-1 CTL suppressed by PD-L1 (Fig 3F). Phos-tag SDS–PAGE is useful for separating a phosphorylated protein from its unphosphorylated counterpart by slower moving rate (Nagy et al, 2018). We conducted Phos-tag SDS–PAGE followed by Western analysis on lysate of the above OT-1/EG-7 co-culture and found that MB potently and dose-dependently enhanced phosphorylation of CD28 in OT-1 CTL (Fig 3G). We further showed that MB treatment enhanced recruitment of P85 by CD28 (Fig 3H), consistent with earlier report that activated CD28 recruited P85 (Tian et al, 2015). Moreover, we found that MB treatment resulted

in overall activation of T-cell function at transcriptomic level (Appendix Fig S3N).

In line with our hypothesis that MB enhanced T-cell function through blocking recruitment of SHP2 by PD-1, we consistently found that MB did not significantly enhance luciferase activity in JP-luc when SHP2 was knockout through CRISPR/CAS9 system (designated JP-luc-sgSHP2) or in the presence of SHP099, an SHP2-specific inhibitor of phosphatase (Chen et al, 2016) (Fig EV3C, Appendix Fig S3O). We also consistently found that MB did not significantly enhance proliferation (Appendix Fig S3P and Q) and IL-2 production of CTLs-sgSHP2 or CTLs treated with SHP099 (Appendix Fig S3R) and cytotoxicity of OT-1 CTLs-sgSHP2 or OT-1 CTL treated with SHP099 (Fig EV3D).

Taken our data together, we solidly showed that MB inhibited PD-1 signaling by directly blocking its recruitment of SHP2 and thus enabled effective TCR signaling.

### MB is a specific iPPI for PD-1/SHP2 interaction

SHP2 plays a critical role mediating signaling of various receptor tyrosine kinases (RTKs), either through direct binding in the case of EGFR (Agazie & Hayman, 2003) or through adapter proteins in the case of FGFR (Li et al, 2014). We asked whether MB inhibited interaction between SHP2 and RTKs, which could elicit severe side effect if used in clinic. To this end, we transfected 293T cells with constructs encoding SHP2 and EGFR or FGFR1 and checked the impact of MB on interaction between SHP2 and these RTKs. Co-IP experiment revealed MB had no detectable impact on the interaction between SHP2 and EGFR (Fig EV3E). In a parallel experiment, we also found no detectable impact of MB on binding between SHP2 and FGFR1 (Appendix Fig S3S). To further confirm our co-IP data, we used luciferase complementation assay. We expressed SHP2-N-luc and EGFR-C-luc or FGFR1-C-luc in 293T cells, such that luciferase activity is a readout for interaction between SHP2 and corresponding RTKs and checked the impact of MB on interaction between SHP2 and RTKs through monitoring luciferase activity. Results showed that MB did not inhibit the interaction between SHP2 and EGFR (Appendix Fig S3T) or interaction between SHP2 and FGFR1 (Appendix Fig S3U). Taken together, our data strongly suggested that MB specifically inhibited interaction between SHP2 and PD-1, but not the interaction between SHP2 and RTKs.

### MB shrinks tumor *in vivo* through enhancing cytotoxic function of CTL

PD-1 inhibitors have shown impressive treatment effect in clinic. We went further to test the ability of MB to shrink tumors *in vivo*. To this end, we inoculated EG7-L1 subcutaneously (S.C.) in C57BL/6J mice on day 1 and intravenously (i.v.) injected activated OT-1 CTL on day 3 and 6, respectively, and randomized the mice into 3 groups ($n = 5$ for each group) for treating with vehicle, aPD1 (10 mg/kg, every other day, i.p.), and MB (20 mg/kg/day, i.g.), respectively (Fig 4A). Tumor size was documented every other day by calipering. While tumors in vehicle-treated group continued growing, aPD1 or MB treatment shrank the tumor significantly in a dose-dependent manner (Figs 4B and C, and EV4A). Tumor weight of antibody-treated or MB-treated group was significantly lower than that of vehicle-treated group by the end of treatment (Fig 4D).

This result was repeated using OT-1 donor and host mice of RAG1$^{-/-}$ background (Fig EV4B and C, Appendix Fig S4A–C), arguing that antigen-specific CTLs were the critical elements for shrinking tumors. Histologically, tumors showed high cellularity and tumor cells showed higher nucleus/cytoplasm ratio in vehicle-treated group. Moreover, cells of vehicle-treated tumor frequently took on a spindle-like shape (Appendix Fig S4D, left panel). Occasional apoptotic bodies were noticeable in hematoxylin-and-eosin-stained tissue sections. These histological features suggested malignant nature of tumor cells and tumor nodules in vehicle-treated mice. In stark contrast, tumors were characterized with intra-tumoral spaces and frequent apoptotic bodies in the aPD1- or MB-treated groups, indicative of regression process (Appendix Fig S4D, middle and right panel). CTL infiltration was significantly enhanced in tumors treated with aPD1 or MB (Fig 4E, upper row, Fig 4F). Ki-67 staining revealed that cell proliferation was inhibited in aPD1- or MB-treated tumors in comparison with vehicle-treated group (Fig 4E, lower panel, Fig 4F). To check the functional state of tumor-targeting CTLs specifically, we transplanted OT-1 CTL of CD45.1 background into CD45.2 host, such that tumor-infiltrating CD8$^{+}$ OT-1 T cells could be analyzed through FACS by gating on CD45.1-positive population (Appendix Fig S4E). We found that MB treatment dramatically enhanced expression of perforin, granzyme B (GZMB), and IL-2 by CTL in tumor nodules (Fig 4G and H). In line with our hypothesis that MB activated T-cell function through inhibiting PD-1 signaling, we found that PD-1KO OT-1 CTL on its own shrank tumor allografts and that MB treatment did not further significantly shrink the tumor volume (Fig EV4D, Appendix Fig S4F).

We went further to model cancer patients with transgenic mouse models of autochthonous lung cancer. We generated TetO-EGFR L858R/CC10rtTA bitransgenic mice (designated EC mice) following our earlier protocol (Ji *et al*, 2006). EC mice developed lung adenocarcinoma after being fed with doxycycline-containing diet for 2 weeks (Appendix Fig S4G–I). Clinical data have shown that PD-1 antibodies are not effective in treating EGFR mutation-positive lung cancer patients (Yoshida *et al*, 2018). Consistently, we found that aPD1 administration showed no noticeable treatment effect on EC tumors (Appendix Fig S4J and K). In striking contrast, we found that

MB treatment led to dramatic regression of EC tumors (Fig 4I–K). Depleting CD8$^{+}$ T cells with aCD8 antibody (Appendix Fig S4L) largely eliminated the tumor shrinking effect by MB (Fig 4J–M), indicating that CD8$^{+}$ T cells played a critical role in mediating the treatment effect of MB. FACS analysis revealed that tumor-infiltrating CD8$^{+}$ T cells in the MB treatment group secreted more IL-2, granzyme B, and perforin (Fig EV4E). Of note, MB was not toxic to EGFR mutant lung cancer cells at concentrations reached at serum level and shows limited toxicity to these cells at super-physiological concentrations (Appendix Fig S4M and N).

Together, our data showed that MB shrank the tumor through activating CD8 cytotoxic T cells *in vivo*.

### MB effectively activates human CD8 T cells

We further asked whether MB was effective in enhancing human T-cell function. To this end, we isolated human peripheral blood mononuclear cells from healthy donors for CFSE dilution assay and quantified the impact of MB on the proliferation of CTLs in the presence of PD-L1 through FACS analysis by gating on CD8$^{+}$ populations. In this experiment, we compared the PD-1 inhibiting activity of MB with pembrolizumab and nivolumab, 2 most popular PD-1 antibodies currently used in clinic. Result showed that in the presence of PD-L1, 100 nM of MB elicited stronger proliferation of CD8$^{+}$ T cells than 25 μg/ml of pembrolizumab or 20 μg/ml nivolumab, respectively (serum concentration achievable in patients for pembrolizumab (Patnaik *et al*, 2015) and nivolumab (Brahmer *et al*, 2010), respectively) (Figs 5A and EV5A, Appendix Fig S5A). Consistently, MB effectively counteracted PD-L1 on suppression of IL-2, IFNγ, perforin, and granzyme B expression in CD8$^{+}$ T cells (Figs 5B and EV5B–E).

We also quantitatively measured ability of MB to counteract the inhibition imposed by PD-L1 on T cells using JP-luc and superantigen SEE-loaded Raji-L1. Based on this system, 100 nM of MB stimulated similar luciferase activity to 25 μg/ml of pembrolizumab or 20 μg/ml of nivolumab. 1 μM or larger amount of MB stimulated stronger luciferase activity than those by antibodies (Fig 5C). These data solidly showed that MB effectively activated human CD8$^{+}$ T cells.

---

**Figure 4. MB shrinks tumor *in vivo* through enhancing cytotoxic function of CTL.**

A   Schematic of the xenograft mouse model for MB treatment. C57BL/6J mice were inoculated with EG7-L1 cells (2 × 10$^6$ cells, s.c.) on the right flank on day 1, followed by injection (2 × 10$^6$ cells, i.v.) of CD45.1$^{+}$ CTL on day 3 and 6, respectively. The mice were randomized into three groups (*n* = 5) and treated with vehicle, MB (20 mg/kg/day, i.g.), and aPD1 (10 mg/kg, every other day, i.*p.*) served as positive control. Tumor size was measured through calipering every other day. EG7-L1: EG7 overexpressing PD-L1.

B   Tumor growth curve for (A) was shown as mean ± SEM.

C   Representative image of the tumor on day 10 mentioned in (A).

D   Tumor weight on day 10 (*n* = 5) mentioned in (A).

E   Immunohistochemical analysis of cell proliferation marker ki-67 and CD8 in tumor nodules on day 10 (*n* = 5) in (A).

F   Statistical results of (E).

G   Expression of perforin, GZMB, and IL-2 by CD8$^{+}$ CD45.1$^{+}$ TILs was analyzed by flow cytometry.

H   Statistical results of (G).

I   Schematic of EGFR L858R mice treated with MB.

J   Tumor burdens recorded through computed tomography (CT) scanning for EGFR L858R mice (*n* = 5, lung section = 3). Mice were treated with MB (20 mg/kg/day, i.g.) for 1 week, followed by imaged with CT again to record tumor burden.

K   Statistical results of (J).

L   Hematoxylin and eosin staining of lung section of EGFR L858R mice (*n* = 5, lung section = 3) in (J).

M   Statistical results of (L).

Data information: Data are representative of three independent experiments and were analyzed by unpaired *t*-test. Error bars denote SEM. **P* < 0.05; ***P* < 0.01; ****P* < 0.001; *****P* < 0.0001.

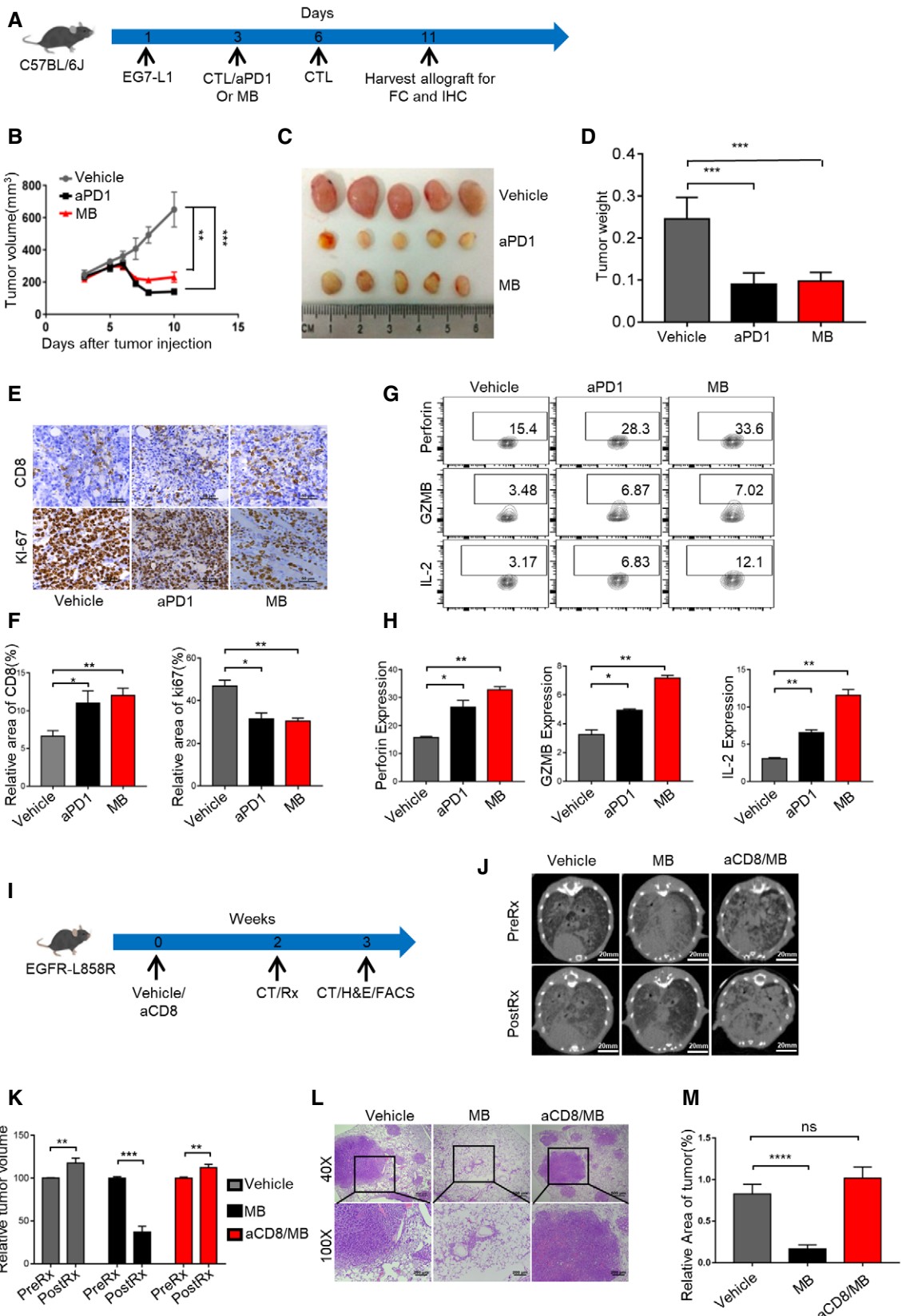

Figure 4.

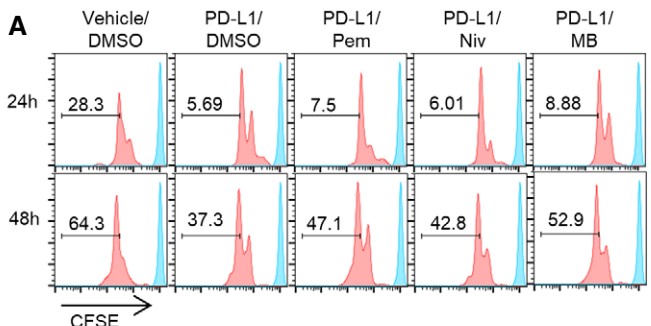

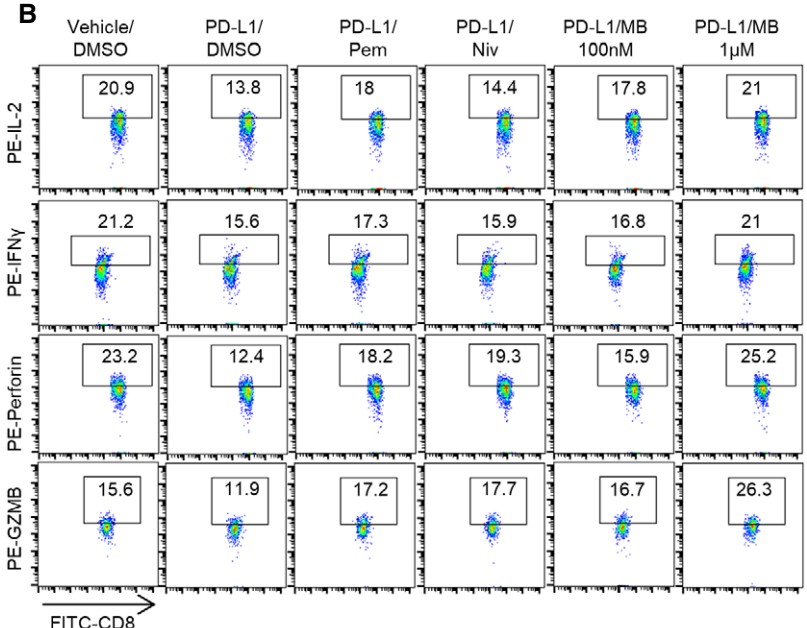

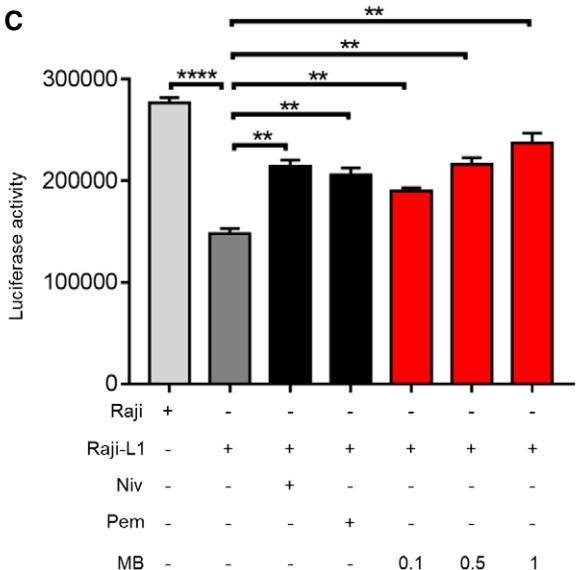

**Figure 5.**

◀

**Figure 5.  MB is effective to activate human CD8+ T cells.**

A   Effect of MB on proliferation of human CD8+ T cells. CFSE-labeled human peripheral blood mononuclear cells (PBMC) were preincubated with DMSO, MB, 25 µg/ml of pembrolizumab, or 20 µg/ml of nivolumab for 1 h and then seeded in the 96-well plate precoated with 10 µg/ml of aCD3/aCD28 in the presence of 10 µg/ml human PD-L1 protein. Cell proliferation was analyzed through FACS analysis of dilution of CFSE.

B   Cytokine and cytolytic granule production in CD8+ T cells. Human peripheral blood mononuclear cells from healthy donors were stimulated with 5 µg/ml of phytohaemagglutinin for 2 days and then rested for 1 day. Cells were pretreated with DMSO, MB, 25 µg/ml of pembrolizumab (Pem), or 20 µg/ml of nivolumab (Niv) for 1 h and then seeded in a 96-well plate precoated with 10 µg/ml aCD3/aCD28, with media supplemented with 10 µg/ml of human PD-L1 protein. Cytokine and cytolytic granule production was determined by intracellular flow cytometry by gating on CD8+ populations.

C   Comparison of the efficacy of MB with PD1 inhibitors currently used in clinic to activate T cell. JP-luc cells were stimulated with Raji-L1 or parental Raji cells preloaded with 1 µg/ml of superantigen (SEE) in the presence of MB at indicated concentration or 25 µg/ml of pembrolizumab (Pem) or 20 µg/ml of nivolumab (Niv) for 6 h. Luciferase activity was measured by luminometer. Data are representative of three independent experiments and were analyzed by unpaired *t*-test. JP-luc: Jurkat cell harboring NFAT-luciferase transgene and overexpressing PD-1; Raji-L1: Raji overexpressing PD-L1.

Data information: Data are representative of three independent experiments and were analyzed by unpaired *t*-test. Error bars denote SEM. **$P < 0.01$; ****$P < 0.0001$.

## Discussion

Small molecular PD-1 inhibitors are urgently needed in immuno-oncology clinic. Small molecular drugs outperform antibody drugs with higher penetrating abilities into the interstices of a tumor and targeting intracellular signaling proteins. The procedure for synthesis, purification, and molecular characterization of small molecules guarantees their identity and purity, ensuring them easy to reproduce. In addition, developing effective chemical PD-1 inhibitors will not only lower cost, but free patients from intravenous injections. Indeed, MB exhibited superior treatment effect on EGFR L858R-driven lung cancers in our transgenic mouse model where PD-1 antibody failed to show treatment effect. Likewise, CD8+ T cells in the tumors of MB-treated EG7-L1 allograft tumor model expressed higher levels of effector molecules than those in PD-1 antibody-treated mice. Small molecular drug is therefore a highly promising field for developing PD-1 inhibitors.

Earlier, targeting interaction between PD-1 and PD-L1 has been tried (Zak *et al*, 2016), including CA-170 (Bojadzic & Buchwald, 2018), BMS-1001 and BMS-1166 (Skalniak *et al*, 2017), and peptides (Chang *et al*, 2015; Li *et al*, 2018). The clinical performance of these drugs remains to be determined. More studies are needed to determine whether other steps along the PD-1 signaling pathway could be targeted by small molecules. Our current work identified MB as a potent PD-1 inhibitor through a high-throughput screening. Mechanistic study revealed that MB inhibited PD-1 signaling by blocking recruitment of SHP2 by PD-1. Moreover, MB is effective to enhance cytotoxicity of CTLs *in vitro* and *in vivo*. Our work, therefore, not only characterized a potent small molecular PD-1 inhibitor of substantial possibility to benefit cancer patients immediately (for it is an FDA-approved drug), but presents a novel strategy to develop PD-1 inhibitors. This novel strategy will undoubtedly broaden our choice for developing PD-1 inhibitors.

MB is used to treat patients with methemoglobin levels greater than 30% or those who have symptoms despite oxygen therapy (Committee, 2015). As a drug with a long clinical history, MB is famous for its favorable safety profile. In a 2-year-treatment toxicity study, researchers found that MB-treated mice survived longer than vehicle-treated group (Hejtmancik *et al*, 2002). Other studies have also shown that MB exhibited favorable clinical safety profile (Baddeley *et al*, 2015). Given the facts that the serum concentration reached in patients is around 6 µM (Baddeley *et al*, 2015) and that IC50 to effectively enhance cytotoxicity of CTL is well below micromolar, it is a tempting hypothesis that MB be a highly effective PD-1

inhibitor in clinic for treating cancer patients either by itself or in combination with other drugs. MB is, therefore, an attractive candidate for clinical trial for cancer and other related disease.

Our work showed that MB inhibited SHP2 to function downstream of PD-1 in CTLs in response to PD-L1 stimulation. SHP2 phosphatase inhibitors are expected to possibly enhance CTL function based on the fact that SHP2 negatively impacts on TCR signaling.

We noticed the difference in baseline functions of sgSHP2- versus SHP099-treated T cells. For examples, when co-cultured with SEE-loaded Raji cells, SHP099-treated JP-luc expressed higher luciferase activity than JP-luc-sgSHP2; when stimulated with aCD3/CD28-coated plate, CTL treated with SHP099 proliferated more than CTL-sgSHP2; SHP099-treated CTL expressed higher amount of IL-2 than CTL-sgSHP2; and SHP099-treated OT-1 CTL kills EG7-L1 more efficiently than OT-1-sgSHP2. Recently, Rota et.al. reported that SHP2 was apparently dispensable PD-1 signaling *in vivo* (Rota *et al*, 2018). How could we reconcile this inconsistency? PD-1 predominantly recruits SHP-2 (Yokosuka *et al*, 2012), with details for signaling worked out very recently (Marasco *et al*, 2020). In the case of SHP2 knockout, PD-1 recruits SHP-1 and remains functional (Celis-Gutierrez *et al*, 2019). In line with this rationale, SHP2-deficient T cells could behave similarly to WT T cells. It could well be that SHP099 inhibited SHP2 protein prevents the compensatory recruitment of SHP1 by PD-1; thus, TCR signaling is more robust.

In summary, we identified an FDA-approved drug as a potent PD-1 inhibitor. Equally important, we showed that targeting PD-1/SHP2 interaction was a realistic strategy for developing PD-1 inhibitors.

## Materials and Methods

### Plasmid constructs, transfection, and viral packaging

pCDH-NFAT-luc for delivering 3×NFAT binding sequence-controlled luciferase fragment through virus infection: Backbone of pCDH (System Biosciences, CD500B-1) was digested with SpeI and NotI. DNA fragment featuring luciferase controlled by 3 repeats of NFAT binding sequence (designated NFAT-luc) was synthesized. pCDH backbone and NFAT-luc fragment were assembled by T4 ligase.

pCAGiN-hPD-1/hPD-L1 (human PD-1, human PD-L1) for high expression of hPD-1 and hPD-L1. pCAGiN (Addgene, #13461) was

digested with EcoRI and ClaI. EcoRI and ClaI flanked hPD-1 and hPD-L1 were synthesized. Digested hPD-1 and hPD-L1 were subcloned into identically digested pCAGiN, respectively.

pcDNA3.1-hPD-1-EGFP and pcDNA3.1-hSHP2-mCherry (human SHP2) for colocalization assay: Fragments hPD-1-EGFP and hSHP2-mCherry were synthesized. pcDNA3.1 was digested with Knp1 and Asc1. hPD-1-EGFP and hSHP2-mCherry fragments were identically digested for cloning into pcDNA3.1.

pWPI-mPDL1 (mouse-PDL1) for overexpression of mPD-L1 and IRES-EGFP element in the vector for FACS sorting virus-infected cells: pWPI (Addgene, #12254) and mPD-L1 fragment was digested with SwaI and AscI and joined by T4 ligase.

pCAGiN-hPD-1-C-luciferase for overexpressing fusion gene containing hPD-1 followed by C-terminal half of luciferase. HindIII and Asc1 double digestion modified hPD-1 and pCAGiN-C-luciferase were joined by T4 ligase.

pCAGiP-N-luc-hSHP2 for overexpressing fusion gene containing N-terminal half of luciferase followed by hSHP2 gene fragment. The pCAGiP vector is digested with EcoRI/CIP and blunted with Klenow. N-luc-SHP2 fragment and pCAGiP were joined by T4 ligase.

Lentivirus packaging: co-transfecting pCDH-NFAT-luc, psPAX2, and PMD2.G (Addgene, deposited by Didier Trono, #12260 and #12259) into HEK293T cells using transfection reagent VigoFect (Vigorous Biotechnology, Beijing, China). The culture was replaced with fresh media 6–8 h after the transfection. The supernatant was harvested 48 h after media change and sterilized through a 0.45 μm filter. The recombinant virus stock was stored at −80°C until use.

Delivering plasmid into cells through electroporation: Instructions of Neon® transfection system (Thermo, MPK5000) was strictly followed. Briefly, incubating cells in a single well of 6-well plates with 2 ml of culture medium containing serum and supplements without antibiotics in a humidified incubator at 37°C and 5% $CO_2$. Pelleting cells (Jurkat E6-1 or Raji cells) by centrifugation at 100–400× *g* for 5 min at room temperature (RT). Washing cells with PBS (without $Ca^{2+}$ and $Mg^{2+}$) and resuspending in Resuspension Buffer R at a final density of $2.0 × 10^7$ cells/ml. Gently pipetting the cells to obtain a single cell suspension. Mix 10 μg plasmid DNA with 100 μl cells ($2.0 × 10^7$ cells/ml) in Resuspension Buffer R at RT and electroporating at 1,350 v, 10 ms, 3 pulses for Jurkat E6-1 cells or 1,300 v, 30 ms, 1 pulse for Raji. Slowly removing the Neon® Pipette from the Neon® Pipette Station and immediately transferring the samples into the prepared culture plate containing prewarmed medium.

The gRNA targeting sequences used in this study were as follows:

Human PD-1-gRNA: GGCCAGGATGGTTCTTAGGT (Ren *et al*, 2017);

Human SHP2-gRNA: CTGGACCAACTCAGCCAAAG;

Mouse SHP2-gRNA: GAGGAACATGACATCGCGG (Ruess *et al*, 2018).

## Cell culture and cell lines

All cell lines were maintained in standard tissue culture incubators at 37°C and 5% $CO_2$. Jurkat E6-1 and Raji cells were cultured in RPMI-1640 supplemented with 10% fetal bovine serum (FBS, Gibco, 42G9274K) and 1% penicillin/streptomycin/glutamine (Gibco, 15140122). 293T cells were cultured in DMEM supplemented with 10% calf serum and 1% penicillin/streptomycin/glutamine.

We used the recombinant pWPI-mPDL1 lentivirus to infect EG7 cells for 24 h, replaced with fresh media, and sorted the mPD-L1+ cells by flow cytometer with mPDL1-PE (1:100, eBioscience, 12-5982-81) to derive mPD-L1 expressing EG7 stable cell line (designated EG7-L1). We transfected pCAGiN-hPD1 and pCAGiN-hPDL1 into Jurkat E6-1 and Raji cells, respectively. hPD- and hPD-L1-positive cells were sorted with hPD1-FITC (1:100, BioLegend, 329904) and hPDL1-PE (1:100, BioLegend, 329706) and seeded at single cell/ well in 96-well plate to derive stable cell clones (Jurkat-hPD-1 T-cell line and Raji-L1, respectively). Then, Jurkat-hPD-1 T-cell line infected with recombinant pCDH-NFAT-luciferase lentivirus to generate Jurkat-hPD-1-NFAT-luciferase cell line (designated JP-luc). Jurkat cells transiently co-expressed hPD-1-EGFP and hSHP2-mCherry were generated by electroporating both plasmids into Jurkat cells.

## Antibodies and flow cytometry

The single cell suspension was prepared by mechanic grinding of indicated tissues. For surface marker staining, cells were resuspended at $1 × 10^6$ per 100 μl PBS and stain cells with 1ul of antibody solution. Namely, CD8 (1:100, BD Horizon, 564297), CD45.1 (1:100, BioLegend, 110706), CD45 (1:100, BD Horizon, 557659), and CD8 (1:100, BioLegend, 100738) were used to stain cells at 4°C for 30 min in dark. Intracellular staining kit (eBioscience, 00-5523-00) was used for staining intracellular target proteins following the instructions provided by manufacturer. Cells were resuspended with 500 μl of 1× Fix buffer and kept at room temperature in dark for 30 min or at 4°C overnight. These cells were then washed with 500 μl of 1× permeabilization wash buffer and centrifuged at 395 *g* for 5 min. Cell pellets were resuspended with 100 μl of 1× permeabilization wash buffer. Then, add 1 μl antibodies solution for staining perforin (1:100, eBioscience, 17-9392-80), IL-2 (1:100, eBioscience, 12-7021-82), or GZMB (1:100, BioLegend, 515408) by incubating at room temperature for 45 min in dark. Stained cells were washed with 1 ml of 1× permeabilization buffer before analysis by FACS.

## Immunohistochemistry analysis

Xenograft and lung tissues were fixed with 10% neutral buffered formaldehyde overnight. Paraffin sections were stained with hematoxylin and eosin or subjected to immunohistochemistry for CD8 (1:50, Cell Signaling Technology, 98941) or ki-67 (1:500, Abcam, ab15580).

## Measurement of OT-1 CD8+ T-cell cytotoxicity

Splenocytes isolated from OT-I mice were stimulated with $OVA_{257-264}$ for 3 days in the presence of 10 ng/ml of IL-2 to generate mature CTLs. Cells were centrifuged and cultured in fresh medium containing 10 ng/ml of IL-2 for 2 more days. To measure CD8+ T-cell cytotoxicity, we mixed CTLs and CFSE (eBioscience, 65-0850-84)-labeled EG7-L1 cells in the presence of MB at indicated concentrations ($1 × 10^4$) in the killing medium (LDH: phenol-free RPMI 1640, 2% FBS; FACS with PI or DAPI: RPMI 1640, 10% FBS) at the effect to target ratios of 2:1, 5:1, and 10:1, respectively. After 4 h, the cytotoxic efficiency was measured by quantifying the lactate dehydrogenase (LDH) in media using a CytoTox 96 Non-Radioactive

Cytotoxicity kit (Promega, G1780). Alternatively, apoptotic EG7-L1 cells were stained with PI (10 µg/ml) or DAPI (5 µg/ml) and analyzed by flow cytometry by gating on CFSE/PI or CFSE/DAPI double-positive populations.

### Measurement of cytokine production by OT-I CTL cells

CTLs were cultured and pretreated with protein transport inhibitor (PTI) and DMSO for 1 h at 37°C and 5% $CO_2$ before incubating with CFSE-labeled EG7-L1 cells for 6 h. Cells were fixed with 4% paraformaldehyde (PFA) and permeabilized with saponin (Sigma, 47036) and stained with IL-2-PE (1:100, eBioscience, 12-7021-82), IFNγ-PE-Cy7 (1:100, eBioscience, 25-7311-82), perforin–APC (1:100, eBioscience, 17-9392-80), or GZMB-Alexa Fluro (1:100, BioLegend, 515405). Cytokine production was measured by flow cytometric analysis gating on CFSE negative population.

### CD8[+] T-cell proliferation

In Fig 2A, splenocytes isolated from OT-I mice were labeled with 5 µM of CFSE and seeded in a 96-well plate precoated with 10 µg/ml of aCD3/aCD28 and with culture media supplemented with mPD-L1 protein (Fc tag, Sino Biological, 50010-M03H-10) in the presence of 100 nM of MB or 10 µg/ml of aPD1 antibody. Cells were stained with CD8-PE (1:100, eBioscience, 12-0081-81). CTL proliferation was measured by CFSE dilution through FACS by gating on CD8-positive population.

In Fig 5A, human peripheral blood mononuclear cells from healthy donors were labeled with CFSE and seeded in a 96-well plate precoated with 10 µg/ml of aCD3/aCD28 and with culture media supplemented with 10 µg/ml hPD-L1 protein (Fc tag, Sino Biological, 10084-H02H-100) in the presence of 100 nM of MB or 25 µg/ml of pembrolizumab or 20 µg/ml nivolumab (kindly gifted by Dr. Li Zhang, SYSUCC). Cells were stained with CD8-PE (1:100, eBioscience, MHCD0804) after 24 and 48 h of incubation. CTL proliferation was measured by CFSE dilution through FACS by gating on CD8 positive population.

### Measurement of cytokine production by human CD8[+] T cell

Human peripheral blood mononuclear cells from healthy donators were stimulated with 5 µg/ml phytohaemagglutinin (Sigma) for 2 days and then rested for 1 day. Cells were pretreated with DMSO or MB or 25 µg/ml pembrolizumab or 20 µg/ml nivolumab for 1 h and then seeded in a 96-well plate precoated with 10 µg/ml of aCD3/aCD28, with media supplemented with 10 µg/ml of hPD-L1 protein. Cells were fixed with 4% paraformaldehyde (PFA) and permeabilized with saponin (sigma, 47036). Cells were stained with CD8-FITC (1:100, eBioscience, 11-0081-82), IL-2-PE (1:100, BioLegend, 500306), IFNγ-PE (1:100, BioLegend, 502508), perforin-PE (1:100, BioLegend, 353303), or GZMB-Alexa Fluro (1:100, BioLegend, 515405), and cytokine production was measured by flow cytometric analysis by gating on CD8-positive population.

### Jurkat stimulation assay

For Jurkat cell stimulation assay shown in Fig 2E and F, Appendix Fig S1C and Fig 5C, Jurkat-PD-1 cell transfected with

NFAT-luciferase reporter (designated JP-luc) was generated. Raji stably expressing hPD-L1 was generated (designated Raji-L1) as antigen-presenting cells (APC). Raji cells or Raji-L1 cells were preincubated with 1 µg/ml of SEE superantigen (Toxin Technology, ET404) in serum-free RPMI medium for 1 h at 37°C. JP-luc cells were pretreated with DMSO, MB, or aPD1 antibody for 1 h before mixing with SEE-loaded Raji or Raji-L1 cells at ratio of 1:1 in a 96-well plate for 6 h. Luciferase activity was measured with a luminometer following the manufacturer's instructions (Promega, E2620).

For Fig 2H, JP-luc cells were co-cultured with Raji-L1 or parental Raji cells in a 96-well plate precoated with 10 µg/ml of aCD3/aCD28 for 6 h and the supernatants were collected after 24 h. IL-2 was quantified by ELISA using Human IL-2 ELISA kit (Mlbio, HX2390).

### Quantification of protein–protein interaction

For experiments described in Fig 3D, we coated the wells of a 96-well microtiter plate with 4 µg/ml of Y248-phosphorylated ITSM version of PD-1 peptide (PEPPVPCVPEQTE(p-Y)ATIVFPSGMGTS-SPAR, synthesized by GL Biochem Ltd.) in carbonate buffer by incubating overnight at 4°C or 2–6 h at 37°C following instructions provided by Peptide Coating Kit (Takara, MK100). After blocking the wells with 5% BSA in PBS, recombinant hSHP2-FC (500 ng/ml, R&D Systems, 1894-SH-100) protein was added in the presence of phosphatase inhibitor cocktail and plates were incubated for 1 h at RT. Horseradish peroxidase (HRP)-conjugated anti-FC Tag antibody (1:5,000, Sino Biological, SSA001) was added in each well for 1 h, and binding was quantified using a DuoSet® Ancillary Reagent Kit (R&D Systems, DY008).

For experiments described in figure Appendix Fig S3B, 96-well microtiter plates were coated with Fc-tagged hPD-L1 protein (10 µg/ml) or Fc-tagged mPD-L1 protein (10 µg/ml) overnight at 4°C. After blocking, recombinant 6×His-tagged hPD-1 (500 ng/ml, Sino Biological, 10377-H08H-50) or 6×His-tagged mPD-1 protein (500 ng/ml, Sino Biological, 50124-M08H-100) was added and plates were incubated for 1 h at RT. Horseradish peroxidase (HRP)-conjugated anti-His antibody (1:5,000, BioLegend, 652504) was added in each well for 1 h, and binding was quantified using a DuoSet® Ancillary Reagent Kit.

For experiments described in Fig EV3A and Appendix Fig S3A, Jurkat-hPD-1 T-cell line (parental Jurkat cells served as negative control) was incubated with 1 µg/ml of hPD-L1 protein (Fc-tagged) in the presence of MB at indicated concentration at 4°C for 30 min. Cells were stained with PE-conjugated anti-hIgG-Fc antibody at 4°C for 1 h. Binding of hPD-L1 on cells was determined by flow cytometry.

### Bimolecular fluorescence complementation assay

Jurkat cells were co-transfected with hPD1-C-Luc or PD1[Y248F]-C-Luc and hSHP2-N-luc or hCD28-C-Luc or hCD28[Y191F]-C-Luc and hP85-N-luc. After 24 h of rest, transfected cells were seeded in a 96-well plate precoated with 10 µg/ml of aCD3/aCD28, with media with 10 µg/ml of hPD-L1 protein in the presence of MB at indicated concentrations for 6 h followed by analysis of luciferase activity.

For experiments described in figure Appendix Fig S3T and U, 293T cells were co-transfected with plasmids encoding human

EGFR-C-Luc or FGFR1-C-Luc and hSHP2-N-luc. After 36 h, the transfected cells were seeded in a 96-well plate, treated with MB at indicated concentrations for 6 h followed by analysis of luciferase activity.

## Confocal microscope analysis

Jurkat cells were co-transfected with hPD-1-EGFP and hSHP2-mCherry. After resting for 24 h, transfected cells were seeded on Poly-L-lysine-treated coverslips in 24-well plate. Cells were then incubated with 10 µg/ml of hPD-L1 protein or 2 µM of pervanadate (PVD) for 5 min in the presence of 100 nM or 1 µM of MB for 2 min before fixing with 4% paraformaldehyde and staining with DAPI. Colocalization signal was examined with confocal microphotograph (bar = 2 µm).

## Co-immunoprecipitation (co-IP)

For experiments described in Fig 3C, 293T cells were co-transfected with constructs pCAGiN-PD-1 and pCDNA3.1-SHP2-FLAG for over-expressing human PD-1 and human SHP2 (3 × FLAG-tagged). Cells were incubated with 100 nM, 500 nM, or 1 µM of MB for 6 h, with the last 5 min treated with 2 µM of PVD. Cells were harvested for preparing the protein lysate using NP40 lysis buffer (125 mM Tris–HCl, pH 7.4, 150 mM NaCl, 1% NP40, 1 mM EDTA, 5% glycerol, 1 mM PMSF, supplemented with Roche protease and PhosSTOP phosphatase inhibitor cocktail). The lysates were subjected to co-immunoprecipitation assay with anti-FLAG antibody (1:1,000, SIGMA, F1804). Equal fractions of the immunoprecipitates were subjected to SDS–PAGE and blotted with PD-1 antibody (1:1,000, Cell Signaling Technology, 86163).

For experiments described in Fig 3E and Appendix Fig S3S, 293T cells were co-transfected with human EGFR-C-Luc and SHP2-3 × FLAG or FGFR1-3 × FLAG and SHP2-HA. After 36 h, transfected cells were serum-starved for 12 h and then treated with complete medium and MB for 2 h. Cells were harvested for preparing the protein lysate using NP40 lysis buffer. The lysates were subjected to co-immunoprecipitation assay with anti-FLAG antibody. Equal fractions of the immunoprecipitates were subjected to SDS–PAGE and blotted with anti-EGFR antibody (1:1,000, Abcam, E234,) or anti-HA antibody (1:1,000, Santa Cruz, C1091).

## Western blotting analysis of TCR signaling protein components

For experiments described in Fig 3E, mature OT-1 CTLs were pretreated with 100 nM of MB. These cells were then seeded in 6-well plate precoated with 10 µg/ml of aCD3/aCD28, with media supplemented with 20 µg/ml of mPD-L1 protein and 100 nM of MB for 2 min. Cells were harvested for preparing protein lysate using NP40 lysis buffer. Equal fractions of the lysates were subjected to SDS–PAGE and blotted with Phospho-Zap-70 (1:1,000, Cell Signaling Technology, 2717), ZAP70 (1:1,000, Cell Signaling Technology, 3165), Phospho-PKCθ (1:1,000, Cell Signaling Technology, 9377), PKCθ (1:1,000, Cell Signaling Technology, 13643), Phospho-PLCγ1(1:1,000, Cell Signaling Technology, 14008), PLCγ1(Cell Signaling Technology, 5690), Phospho-AKT (1:1,000, Cell Signaling Technology, 4060S) and AKT (1:1,000, Cell Signaling Technology 9272S).

## Animal experiments

C57BL/6J mice were purchased from Beijing Vital River Laboratory Animal Technology Co. Ltd.. CD45.1[+] OT-1 transgenic mice of C57BL/6J background, Rag1[−/−] mice, and TetO-EGFR L858R/CC10rtTA mice were kept in Institute of Laboratory Animal Science, Jinan University. All wild-type and transgenic mice were age- and sex-matched. All animals were housed in specific pathogen-free conditions and breeding, and all animal procedures were conducted in strict accordance with guidelines for the care and use of laboratory approved by the Institute of Laboratory Animal Science, Jinan University.

C57BL/6J female mice (8 weeks) or Rag1[−/−] female mice (8 weeks) were injected with $2 \times 10^6$ EG7-L1 (s.c.) on the right flank. After tumor growth up to $7 \times 6 \times 6$ mm$^3$ or after indicated days postinoculation, mice were transferred with $2 \times 10^6$ OVA stimulated CD45.1[+] CTL cells (i.v.). Mice were randomized into 3 groups (n=5) for treatment with vehicle, 10 mg/kg of aPD-1 every other day (BioXcell, BP0033-2) (i.p.), and 20 mg/kg/day (i.g.) of MB (TargetMol, 7220-79-3). The tumor growth and tumor weight were appropriately recorded.

EGFR L858R transgenic female mice (8 weeks) were generated by microinjection and characterized. Lung cancer-bearing mice were randomized into 3 groups ($n = 5$) for treating with vehicle, MB, and aCD8/MB, respectively. Anti-CD8a (BioXcell, BE0061) depleting antibody was injected (i.p.) at 200 mg per mouse every other day for 2 weeks before combinational treatment with MB. Tumor burdens were recorded through computed tomography scanning. CT images (recorded by Pingseng Healthcare recorder SNC-100) were analyzed blindly. Mice were sacrificed after the treatment for 1 weeks. Lungs of these mice were prepared for pathological or flow cytometry analysis.

## Human peripheral blood mononuclear cell proliferation

Protocol was approved by ethnic committee of Jinan University. Informed consent was obtained from all subjects, and the experiments were conformed to the principles set out in the WMA Declaration of Helsinki and the Department of Health and Human Services Belmont Report.

## Appendix data information

Exact P-values and statistical tests are listed in Appendix Table S1. Dilutions for all the antibodies are listed in Appendix Table S2.

**Expanded View** for this article is available online.

## Acknowledgements

Authors are grateful for the staff member of Institute of Laboratory Animal Science, Jinan University for taking intensive care of the transgenic mice. Authors thank Dr. Hongbin Ji for critical reading of this manuscript. This work is supported by NSFC (81672309, 81972778), Guangzhou Research Project of Science and Technology for Citizen Health (201803010124), and Science and Technology Program of Guangdong (2017B020227001) to L.C.; NSFC (31800723) and the Fundamental Research Funds for the Central Universities (21618326) to Q.Z.; and NSFC (31701174) to Q. Huang.

## The paper explained

### Problem

PD-1/PD-L1 inhibitory antibodies have been successfully used in immuno-oncology clinics. However, three major problems remain to be explored: (i) Alternative drugs other than antibody need to be explored. Toxicity has been reported for antibody drugs in clinic. Lines of evidence suggest that the toxicity issues may not be the results of PD-1 inhibition per se, but from Fc part of antibodies. (ii) Most of the current antibody drugs used in clinic inhibit PD-1 function by blocking interaction between PD-1 and its ligands. Other strategies for developing PD-1 inhibitors remain to be explored. (iii) Small molecular PD-1 inhibitors targeting steps other than PD-1/PD-L1 interaction remain to be explored.

### Results

We set up a high-throughput screening system consisting of PD-1 expressing T cell and PD-L1 expressing antigen-presenting cell. Using this system, we screened small chemical library for PD-1 inhibitors and identified methylene blue (MB), an FDA-approved drug, as a potent PD-1 inhibitor. Mechanistically, MB blocked recruitment of SHP2 by PD-1 of T cell in response to stimulation by PD-L1 and thus inhibited PD-1 signaling. MB potently restored proliferation, cytokine expression, and cytotoxicity of cytotoxic T lymphocytes (CTL) in the context of PD-1 signaling. MB effectively shrank tumor allografts or autochthonous lung cancer in vivo. MB was also effective to enhance human CTL function in vitro.

### Impact

We have identified an FDA-approved chemical as a potent PD-1 inhibitor, implicating immediate clinical application. Our data showed that targeting PD-1/SHP2 interaction is a reasonable strategy for developing PD-1 inhibitor, which sheds light on novel strategies to develop PD-1 inhibitors.

## Author contributions

Conception and design: LC. Development of methodology: ZF, YT, JHe, PZ, NL, ZZ, CX, SG, JHu, QZ. Acquisition of data: ZF, YT, JHe, PZ, ZC, LL, CD, JC. Analysis and interpretation of data (e.g., statistical analysis, biostatistics, computational analysis): ZF, YT. Writing, review, and/or revision of the manuscript: LC, PZ, KD. Administrative, technical, or material support (i.e., reporting or organizing data, constructing databases): ZF, YT. Study supervision: LC, PZ.

## Conflict of interest

The authors declare that they have no conflict of interest.

## For more information

(i) Authors' webpage: https://faculty.jnu.edu.cn/smkx/cl/list.htm
(ii) Clinical trials of PD-1 inhibitors: https://clinicaltrials.gov/

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
