## [Review Process File · EMBO Molecular Medicine]

Blocking interaction between SHP2 and PD-1 denotes a novel opportunity for developing PD-1 inhibitors

Zhenzhen Fan, Yahui Tian, Zhipeng Chen, Lu Liu, Qian Zhou, Jingjing He, James Coleman, Changjiang Dong, Nan Li, Junqi Huang, Chenqi Xu, Zhimin Zhang, Song Gao, Penghui Zhou, Ke Ding, Liang Chen

Review timeline:

Submission date:	8th Oct 2019
Editorial Decision:	27th Nov 2019
Revision received:	11th Mar 2020
Editorial Decision:	27th Mar 2020
Revision received:	2nd Apr 2020
Accepted:	8th Apr 2020

Editor: Jingyi Hou

Transaction Report:

1st Editorial Decision

27th Nov 2019

Thank you for the submission of your manuscript to EMBO Molecular Medicine. As you will see from the reports below, the referees find the topic of your study of potential interest. However, they raise substantial concerns on your work, which should be convincingly addressed in a major revision of the present manuscript.

Importantly, additional experiments and controls are required to confirm that the observed anti-tumor effect of MB is indeed through blocking the interaction between PDL1 and Shp2. Further, referee #1 points out that some of the presented data are contradicting existing literature, which needs to be satisfactorily addressed.

Overall it is clear that publication of the manuscript cannot be considered at this stage. I also note that addressing the reviewers concerns in full will be necessary for further considering the manuscript in our journal and this appears to require a lot of additional work and experimentation. I am unsure whether you will be able or willing to address those and return a revised manuscript within the 3 months deadline. On the other hand, given the potential interest of the findings, I would be willing to consider a revised manuscript with the understanding that the referee concerns must be fully addressed and that acceptance of the manuscript would entail a second round of review. I should remind you that it is EMBO Molecular Medicine policy to allow a single round of revision only and that, therefore, acceptance or rejection of the manuscript will depend on the completeness of your responses included in the next, final version of the manuscript. For this reason, and to save you from any frustrations in the end I would strongly advise against returning an incomplete revision and would also understand your decision if you choose to rather seek rapid publication elsewhere at this stage.

***** Reviewer's comments *****

Referee #1 (Comments on Novelty/Model System for Author):

Some of the data in the cell culture assays are confusing and contradict to literature.

Referee #1 (Remarks for Author):

Fan et al. present a series of evidence that methylene blue (MB) promotes T cell function by preventing the inhibitory receptor PD-1 from interacting with the tyrosine phosphatase Shp2. The authors also show that MB can rival or even outperform PD-1 blockade antibodies in mouse tumor models. This is a novel finding with translational potential, especially given that MB has already been approved by FDA. Despite this, I am confused by several elements of the study, as outlined below.

Major concerns:

1. I feel that definitive data showing that MB specifically targets the PD-1 pathway is lacking, because PD-1+ T cells were used in the entire paper. In principle, much of the data in the cell cultures and in vivo can be explained if MB stimulates T cell function independent to the PD-L1/PD-1 pathway. In my opinion, PD-1 KO or Shp2 KO are needed for at least some experiments to demonstrate that MB specifically affects PD-1/Shp2 interaction.
2. Fig. 2H, the authors showed that MB more effectively stimulates IL-2 production than anti-PD-1, but they also detected a similar effect in PD-L1 negative condition. In fact, there was little difference in IL-2 between PD-L1 negative and positive conditions, suggesting that MB effect is independent to PD-L1/PD-1 signaling.
3. Fig. 4, the authors showed MB can promote T cell mediated anti-tumor activity in a lung cancer model which does not respond to aPD-1. Wouldn't this data suggest that MB activates T cells at least partly independent of PD-1 pathway?
4. The authors suggested that MB blocks PD-1/Shp2 interaction. However, the roles of Shp2 in the PD-1 pathway has been questioned by Rota et al., which reported that Shp2 is dispensable for PD-1 function in vivo. This previous finding seems to contradict to the main claim of the manuscript that MB shrinks tumor by blocking PD-1/Shp2 interaction. This concern needs to be addressed, perhaps using Shp2 KO cells. If the authors' model is correct, MB should not have any effect in this background.
5. Fig. 3A, how could a soluble PD-L1 can stimulate PD-1 in Jurkat cells? In common sense, an immunological synapse is needed to exclude CD45 from PD-1, and a soluble ligand would not be able to do so. I'm not going to argue that it is impossible. However, to ensure it is not a fixation artifact, I'd like to see a time-lapse movie of SHP2 recruitment upon the addition of PD-L1. A formal quantitation with statistical test is required.
6. No information on how MB inhibits PD-1/Shp2 interaction, but not EGFR/Shp2 interaction. Does MB binds to PD-1 or Shp2?

Minor concerns:

1. Fig. 3G, why phos-CD28 weaker in the no PD-L1 conditions?
2. Fig. 1B, axis labels need to be fixed.

Referee #2 (Comments on Novelty/Model System for Author):

The authors demonstrate the observed effect in several different model systems using murine and human cells in vitro, but also a murine tumor model in vivo. A variety of readout methods is utilized including FACS, classical Western Blot Analyses, and immunofluorescence. The clinical relevance is clear, as PD-1 PD-L1-blocking antibodies are currently the number one blockbuster in cancer therapy. The serious side effect profile, however obliges a search for alternatives.

Referee #2 (Remarks for Author):

Fan and colleagues invented a very clever screening system for PD-1-related signaling with the intent to find small molecule inhibitors with similar therapeutic effects like the checkpoint-inhibiting antibodies Pembrolizumab and Nivolumab. With this system they identified Methylene Blue (MB), a substance used already in humans for other indications. They could convincingly show, that MB concentrations which could be reached easily within patients' blood were just as efficient as

antibody-mediated inhibition of PD-1. They elucidated the molecular mechanism, showing that PD-1/SHP2 protein/protein-interaction is inhibited, resulting in interruption of the efferent signaling cascade. Their experiments show effectivity of inhibition both in vitro and in vivo. The clinical relevance of these findings is obvious. I have hence very little criticism:

Sometimes the authors use terms which are linguistically not correct, i.e. page 5: "...interactions (PPI) was previously ..." should be: "...interactions (PPI) were previously ...". or on the next page: "... but shaded light on ..." must read: "... but shed light on ..."

On page 10 they use the word "trend". This makes the results weaker than they are, because trend is usually used when data do not reach statistical significance, although this is clearly the case here. I would suggest to use the word "pattern" instead.

There is one issue I must unfortunately address here: In figure 5A, the 3rd and 4th panel in the top row (24h PD-L1/Pem and 24h PD-L1/Niv are identical i.e. copy paste. I assume that was a mistake when generating the figure. This must be corrected.

Referee #3 (Comments on Novelty/Model System for Author):

The overall argument and science in this article seems outstanding and worthy of publication. I don't have the technical skills to review the scientific methods and when I contacted my colleague to get her help, she had sent the manuscript to me because she felt that she didn't have the skill set to review this manuscript. I would say that if the experimental work can be appropriately reviewed and looks okay, then I would publish it on all other counts. The person I think would be able to review this manuscript is Anusha Kalbasi anushakalbasi@mednet.ucla.edu I don't think it is fair to the authors for the paper to not be accepted because of my inability to review it. But I will say that two of us have struggled with it and that may indicate that it is not written at the right level. Likely too much detail is provided on how the authors got to the SHP2 result and the paper needs to be written to be more readable and less of a detailed account of all the experiments that were done. I am not sure though. I truly apologize, especially for the delay.

1st Revision - authors' response

11th Mar 2020

Referee #1 (Remarks for Author):

Fan et al. present a series of evidence that methylene blue (MB) promotes T cell function by preventing the inhibitory receptor PD-1 from interacting with the tyrosine phosphatase Shp2. The authors also show that MB can rival or even outperform PD-1 blockade antibodies in mouse tumor models. This is a novel finding with translational potential, especially given that MB has already been approved by FDA. Despite this, I am confused by several elements of the study, as outlined below.

1. I feel that definitive data showing that MB specifically targets the PD-1 pathway is lacking, because PD-1+ T cells were used in the entire paper. In principle, much of the data in the cell cultures and in vivo can be explained if MB stimulates T cell function independent to the PD-L1/PD-1 pathway. In my opinion, PD-1KO or SHP2KO are needed for at least some experiments to demonstrate that MB specifically affects PD-1/SHP2 interaction.

We thank Reviewer for this great suggestion of using genetically engineered T cells to clearly show that MB enhances T cell activity by inhibiting PD-1 signaling. We therefore tested MB's impact on various genetically engineered T cells through the following experiments:

Experiment#1:

To clarify whether MB exerts its impact on T cells through inhibiting PD-1 function by blocking its recruitment of SHP2, we knockout or overexpressed PD-1 in Jurkat cells harboring NFAT-luciferase (designated J-luc) clones. Specifically, we knockout PD-1 through sgRNA/CAS9 system (Designated J-luc-sgPD-1), overexpressed PD-1 (designated JP-luc), or overexpressed PD-1 but knockout SHP2 (JP-luc-sgSHP2) (supporting Figure 1A-D). We then co-cultured these engineered Jurkat cells with Raji or Raji-PD-L1 and check the impact of MB on luciferase activity, with aPD1 as a positive control.

Our data showed that 1 μ M of MB abolished PD-1's inhibitory effect on NFAT controlled luciferase activity in JP-luc cells. In J-luc-sgPD-1 cells, this effect is largely lost. Similarly, this effect is lost in JP-luc-sgSHP2 cell or JP-luc treated with 10 μ M of SHP099, a small molecular SHP2-selective inhibitor (Fortanet et al. (2016), *J. Med. Chem.*, 59 7773) (supporting Figure 1E).

We have updated these results in **Figure 2F** and **Figure EV2E, EV3Q, EV3R & EV3S** in our manuscript.

Supporting Figure 1: MB activates NFAT binding-sequence controlled luciferase activity in Jurkat cells through inhibiting PD-1 signaling.

A. Sequencing chromatogram of PCR products of human PD-1 locus around the sgRNA target site in Jurkat cells treated with sgRNA/CAS9 lentivirus. **B.** Sequencing chromatogram of PCR products of human SHP2 locus around the sgRNA target site in Jurkat cells treated with sgSHP2/CAS9 lentivirus. **C.** Western blot analysis of PD-1 expression in JP-luc, J-luc-sgPD-1 stable cell line stimulated without or with 500 ng/ml of PHA for 2 days. **D.** Western blot analysis of SHP2 expression in JP-luc and JP-luc-sgSHP2. **E.** Impact of MB on luciferase activity of various engineered Jurkat T cells. The sgRNA targeting sequences used in this study were listed below:

Human PD-1-sgRNA: GGCCAGGATGGTTCTTAGGT;

Human SHP2-sgRNA: CTGGACCAACTCAGCCAAAG.

J-luc: Jurkat cell harboring NFAT-luciferase transgene; J-luc-sgPD-1: J-luc cells treated with lentivirus expressing sgPD-1/CAS9 simultaneously; JP-luc: J-luc cell overexpressing PD-1; JP-luc-sgSHP2: JP-luc cells treated with lentivirus expressing sgSHP2/CAS9. SHP099: JP-luc cells treated with 10 μ M of SHP099.

Data are representative of three independent experiments. Statistics were analyzed by unpaired t-test. Error bars denote s.e.m. *P < 0.05; **P < 0.01; ***P < 0.001; ****P < 0.0001.

Experiment#2:

We tested MB's impact on proliferation of CD8⁺ T cells of wild-type C57BL/J background (WT), PD-1 knockout (PD1KO) (Nishimura, H. et.al. *Int Immunol* 10, 1563-1572 (1998)), SHP2 knockout through lentiviral delivery of sgSHP2/CAS9 (sgSHP2), or T cells treated with SHP099 (supporting Figure 2A-D).

We found that while aCD3/CD28 activated proliferation of WT CD8⁺ T cells, PD-L1 administration inhibited proliferation of these T cell. 1 μ M of MB abolished PD-1's inhibitory effect of proliferation of WT CTLs. Most importantly, this effect was lost on PD-1KO CTLs. We also found that neither aPD-1 nor MB further significantly enhanced proliferation of PD-L1 treated CTLs in presence of SHP099 (supporting Figure 2E). We also found that the ability of aPD-1 or MB to enhance proliferation of PD-L1 treated CTL-sgSHP2 was significantly diminished in comparison to that of WT CTLs.

We have updated these data in **Figure 2A & 2B** and **Figure EV10, EV2P, EV3S & EV3T** in our manuscript.

Supporting Figure 2. MB activates proliferation of CTLs through inhibiting PD-1 signaling.

A. FACS analysis the expression of PD-1 in CTL from PD-1 knockout (PD-1KO) mice or WT mice. **B.** Western blot analysis the expression of PD-1 in CTL from PD-1KO mice or WT mice. **C.** Sequencing chromatogram of PCR products of mouse SHP2 locus around the sgRNA target site. **D.** Western blot analysis of SHP2 expression in WT CTLs or CTLs infected with sgSHP2/CAS9 lentivirus. **E.** Effect of MB on the proliferation of WT CTLs, PD-1KO CTLs, sgSHP2 CTLs and SHP2099. The sgRNA targeting mouse SHP2 sequence used in this study: GAGGAACATGACATCGCGG.

Splenic cells were stained with 5 μ M of CFSE and seeded into aCD3/CD28 coated 96-well plates. 10 μ g/ml of PD-L1 were administered in media. Cell proliferation were checked by monitoring dilution of CFSE 48 hours after CD3/CD28 stimulation by gating on CD8 positive population through FACS analysis.

WT: splenic cell from wildtype C57BL/J mice. PD-1KO: splenic cell from PD-1 knockout mice. sgSHP2: splenic cells infected with sgSHP2/CAS9 lentivirus.

Data are representative of three independent experiments. Statistics were analyzed by unpaired t-test. Error bars denote s.e.m. *P < 0.05; **P < 0.01; ***P < 0.001; ****P < 0.0001.

Experiment#3:

We tested MB's impact on IL-2 expression by CD8⁺ T cell of wild-type background (WT), PD-1 knockout (PD-1KO) (Nishimura, H. et.al. Int Immunol 10, 1563-1572 (1998)), SHP2 knockout T cell (sgSHP2), or T cells treated with SHP099 (supporting Figure 3).

Our data showed that 1 μ M of MB significantly ameliorated PD-1's inhibitory effect on IL-2 expression by WT CTL. However, this effect is lost in case of PD-1KO CTLs. We also found that the ability of MB to enhance IL-2 expression of

CTL-sgSHP2 cells or by CTLs in presence of SHP099 was significantly inhibited in comparison to that of WT CTLs.

We have updated our results in **Figure EV3U** in our manuscript.

Supporting Figure 3. MB enhances IL-2 expression by CTLs through inhibiting PD-1 signaling.

A. FACS analysis of IL-2 expression by CTLs of indicated genetic backgrounds and treatment. **B.** Bar graph of results shown in **A**. The sgRNA targeting mouse SHP2 sequence used in this study was as follows: Mouse SHP2-gRNA: GAGGAACATGACATCGCGG.

Splenic cells were seeded into aCD3/CD28 coated 96-well plates. 10 μ g/ml of PD-L1 were administered in media. IL-2 expression by CTLs was monitored by FACS analysis through intracellular staining. MB was administered at 1 μ M.

WT: wildtype C57BL/6J mice. PD-1KO: PD-1 knockout mice. sgSHP2: WT CTLs infected with sgSHP2/CAS9 lentivirus. SHP099: WT CTLs in the presence of 10 μ M of SHP099.

Data are representative of three independent experiments. Statistics were analyzed by unpaired t-test. Error bars denote s.e.m. * $P < 0.05$; ** $P < 0.01$; *** $P < 0.001$; **** $P < 0.0001$.

Experiment#4:

We tested MB's impact on killing efficacy of OT-1 CTLs of wild-type background (WT), PD-1 knockout (PD-1KO) (Nishimura, H. et.al. Int Immunol 10,

1563-1572 (1998)), SHP2 knockout (sgSHP2), or in the presence of 10 μ M of SHP099 (supporting Figure 4).

Our data showed that 1 μ M of MB abolished PD-1's inhibitory effect on OT-1 CTL to kill EG-7-PD-L1. OT-1 CTLs of PD-1KO background were not sensitive to treatment with PD-1 antibody or MB in terms of killing EG-7 or EG7-PD-L1 target cells. We found similar pattern in case of OT-1 CTL-sgSHP2 or OT-1 CTLs treated with SHP099.

We have updated our result in **Figure 1E & 1F** and **Figure EV10, EV1P & EV3V** in our manuscript.

Supporting Figure 4. MB enhances killing efficacy of OT-1 CTLs through inhibiting PD-1 signaling. **A.** Impact of MB on cytotoxicity of WT OT-I CTLs, PD-1KO, sgSHP2 or OT-1 in the presence of 10 μ M SHP099 against EG7 or EG7- PD-L1. **B.** Bar graph of results shown in A.

sgRNA targeting mouse SHP2 sequence used in this study: GAGGAACATGACATCGCGG.

Data are representative of three independent experiments. Statistics were analyzed by unpaired t-test. Error bars denote s.e.m. *P < 0.05; **P < 0.01; ***P < 0.001; ****P < 0.0001.

Experiment#5:

We tested MB's impact on tumor-shrinking ability of transferred OT-1 CTLs of wild-type background (WT) or PD-1 knockout (PD-1KO) (Nishimura, H. et.al. Int Immunol 10, 1563-1572 (1998)).

We found that EG-7-PD-L1 allograft tumors continued growing after transferred OT-1 CTLs. PD-1 antibody or MB treatment enabled OT-1 CTL to shrink tumor allografts. Interestingly, transferred PD-1KO OT-1 CTLs effectively shrank tumors without aPD1 or MB treatment. Moreover, treatment with PD-1 antibody or MB didn't further enhance the ability of PD-1KO OT-1 cell to shrink tumor allografts (Supporting Figure 5).

We have updated these results in **Figure EV4I, EV4J & EV4K** in our manuscript.

Supporting Figure 5. PD-1 is critical for mediating tumor-shrinking effect of MB in vivo

A. Tumor growth curves of subcutaneous tumor allograft. EG7-PD-L1 cell (2×10^6 cells) were subcutaneously injected into the right flank of C57BL/6J mice ($n = 4$ per group). The mice were then injected with WT or PD-1KO CTLs (i.v.) on day 3 and 6 respectively and treated with vehicle, aPD1 (i.p.10mg/kg, every other day) or MB (i.g. 20 mg/ kg/ day). Tumor volume was shown as mean \pm s.e.m. **B.** Image of the tumors. **C.** Bar graph of tumor weight shown in B. ($n=4$).

Data are representative of three independent experiments. Statistics were analyzed by unpaired t-test. Error bars denote s.e.m. * $P < 0.05$; ** $P < 0.01$; *** $P < 0.001$.

Taken together, these data strongly supported that MB enhanced T cell function and that this effect was mediated by PD-1.

2. Fig. 2H, the authors showed that MB more effectively stimulates IL-2 production than anti-PD-1, but they also detected a similar effect in PD-L1 negative condition. In fact, there was little difference in IL-2 between PD-L1 negative and positive conditions, suggesting that MB effect is independent to PD-L1/PD-1 signaling.

We apologized for our mistakes in preparing Figure 2H. Our experiment was conducted as following: 5×10^4 JP-luc were seeded into 96 well plate. These JP-luc cells were cultured by their own (grey group), co-cultured with parental 5×10^4 Raji cells (black group) or Raji-PD-L1 (red group). The cultured cells were treated with

vehicle, 10 $\mu\text{g/ml}$ of nivolumab PD-1 antibody, or 1 μM MB respectively. IL-2 was quantified in the media to monitor T cell activation (Supporting Figure 6).

We apologized that we mis-labeled the data column when extracting data for making figure. The figure has now been corrected.

Supporting Figure 6. MB enhances IL-2 expression.

5×10^4 JP-luc were seeded into 96 well plate. These JP-luc cells were cultured by their own (grey group), co-cultured with parental 5×10^4 Raji cells (black group) or Raji-PD-L1 (red group). The cultured cells were treated with vehicle, 10 $\mu\text{g/ml}$ of nivolumab PD-1 antibody, or 1 μM MB respectively.

JP-luc: Jurkat cell harboring NFAT-luciferase transgene and overexpressing PD-1.

Data are representative of three independent experiments. Statistics were analyzed by unpaired t-test. Error bars denote s.e.m. * $P < 0.05$; ** $P < 0.01$; *** $P < 0.001$.

3. Fig. 4, the authors showed MB can promote T cell mediated anti-tumor activity in a lung cancer model which does not respond to aPD-1. Wouldn't this data suggest that MB activates T cells at least partly independent of PD-1 pathway?

Ever since Chen and colleagues put forward the concept of treating cancers with PD-1/PD-L1 inhibitors (Cancer Res 65, 1089-1096 (2005)) and conducted the first clinical trial (J Clin Oncol 28, 3167-3175), both antibodies and small molecules are now intensively investigated for PD-1 inhibitors. Small molecules differ from antibody drugs in terms of tissue-penetrating ability, half-life in blood, antibody-dependent cytotoxicity et.al.. A few possibilities might explain superior treatment effect of a small molecular PD-1 inhibitor to an antibody in terms of inhibiting lung cancer. For example, it could be that it's easier for small molecular inhibitors to reach lung tissues than antibodies. Also, antibody's antitumor effect involves a complex interaction between T cell and myeloid cell in local microenvironment (Dahan, R. et al. Cancer Cell 28, 285-295). Possibilities exist that T cells tagged with anti-PD-1 antibody were phagocytosed by local macrophages or other phagocytes. These and other potential possibilities could lead to small molecular inhibitors exhibit stronger treatment effect in vivo than antibodies, although both small molecular inhibitors and antibodies inhibit PD-1 signaling equally well in vitro.

4. The authors suggested that MB blocks PD-1/Shp2 interaction. However, the roles of Shp2 in the PD-1 pathway has been questioned by Rota et al., which reported that Shp2 is dispensable for PD-1 function in vivo. This previous finding seems to contradict to the main claim of the manuscript that MB shrinks tumor by blocking PD-1/Shp2 interaction. This concern needs to be addressed, perhaps using Shp2KO cells. If the authors' model is correct, MB should not have any effect in this background.

We are grateful for Reviewer for suggesting this great experiment. As shown in Supporting Figure 1-4, we did the experiment and found that the ability of MB to enhance luciferase activity of JP-luc-sgSHP2 or JP-luc treated by SHP099 was compromised in comparison to that of untreated WT JP-luc, to enhance proliferation and IL-2 expression of CTL-sgSHP2 or CTL treated with SHP099 compared to those of untreated WT CTLs, and to enhance the cytotoxicity of OT-1 CTL-sgSHP2 and OT-1 CTL treated with SHP099 compared to untreated WT OT-1. We extracted the related data of sgSHP2 or SHP099 treated cells and listed these data in Supporting Figure 7 below.

Of note, we did see difference in baseline functions of sgSHP2 versus SHP099 treated T cells. For example, when co-cultured with SEE-loaded Raji-PD-L1 cells, SHP099-treated JP-luc expressed higher luciferase activity than JP-luc-sgSHP2; when stimulated with plate-coated aCD3/CD28 and PD-L1, higher percentage of CTL-sgSHP2 proliferated than CTL treated with SHP099; SHP099-treated CTL expressed higher amount of IL-2 than CTL-sgSHP2; SHP099-treated OT-1 CTL kill EG7-PD-L1 more efficiently than OT-1-sgSHP2.

We think that the possible explanation lies in ways of PD-1's recruitment of downstream mediators. PD-1 predominantly recruits SHP-2. A very recent reports suggested that in the case of SHP2 knockout, PD-1 recruits SHP-1 and remains functional (Celis-Gutierrez J., Cell Rep. 2019 Jun 11;27(11):3315-3330.e7). In line with this rationale, SHP2 deficient T cells could behave similarly to WT T cells. SHP099 inhibited SHP2 protein prevent the compensatory recruitment of SHP1 by PD-1, thus TCR signaling is more robust.

We are so grateful to Reviewer for suggesting these experiments. **We have embodied these findings and discussed the result in more detail in Discussion part (paragraph 3 in page 20).**

Supporting Figure 7: SHP2 plays an important role mediating the activating function of MB on Jurkat T cells and mouse CTLs.

A. Impact of MB on luciferase activity of various engineered Jurkat T cells. **B.** Effect of MB on the proliferation of WT CTLs, CTLs-sgSHP2 and SHP2099 treated CTLs. **C.** FACS analysis of IL-2 expression by WT CTLs, CTLs-sgSHP2 and SHP2099 treated CTLs. **D.** Impact of MB on cytotoxicity of WT OT-1 CTLs, OT-1-sgSHP2 and SHP2099 treated OT-1 CTLs against EG7 or EG7- PD-L1. sgRNA targeting sequences used in this study were listed below:

Human SHP2-gRNA: CTGGACCAACTCAGCCAAAG;

Mouse SHP2-gRNA: GAGGAACATGACATCGCGG.

JP-luc: Jurkat cell harboring NFAT-luciferase transgene and overexpressing human PD-1; JP-luc-sgSHP2: JP-luc cells treated with lentivirus expressing sgSHP2/CAS9 simultaneously. SHP099: JP-luc cells treated with 10 μ M of SHP099.

Data are representative of three independent experiments. Statistics were analyzed by unpaired t-test. Error bars denote s.e.m. * $P < 0.05$; ** $P < 0.01$; *** $P < 0.001$; **** $P < 0.0001$.

5. Fig. 3A, how could a soluble PD-L1 can stimulate PD-1 in Jurkat cells? In common sense, an immunological synapse is needed to exclude CD45 from PD-1, and a soluble ligand would not be able to do so. I'm not going to argue that it is

impossible. However, to ensure it is not a fixation artifact, I'd like to see a time-lapse movie of SHP2 recruitment upon the addition of PD-L1. formal quantitation with statistical test is required.

As suggested by Reviewer, we co-transfected Jurkat cell with constructs for expressing PD-1-EGFP and SHP2-mCherry. The engineered Jurkat cell were seeded onto poly-lysine coated cover slides. 10 $\mu\text{g/ml}$ of PD-L1 was administered in media. Colocalization of EGFP and mCherry was recorded in a time-lapse recorder LSM880. We observed that EGFP and mCherry began to colocalize 3 minutes post PD-L1 administration and became obvious from minute 4 on (supporting figure 8). We have attached the video for Reviewer's reference.

Supporting Figure 8. PD-L1 induced colocalization of PD-1 and SHP2 in Jurkat cells.

A. PD-1-EGFP and SHP2-mCherry cotransfected Jurkat cell were seeded onto poly-lysine coated cover slides. PD-L1 was administered in media. Confocal microscopy time-lapse imaging was conducted to record colocalization of PD-1 and SHP2. **B, C.** The percentage of cells positive of colocalization signal in the absence or presence of PD-L1.

6. No information on how MB inhibits PD-1/Shp2 interaction, but not EGFR/Shp2 interaction. Does MB binds to PD-1 or Shp2?

We have been pursuing the answer to this question ever since we identified MB as an inhibitor for PD-1/SHP2 interaction 3.5 years ago. Using Biacore™ T200, we detected the binding of MB to SHP2 protein (Supporting Figure 9). In collaborating with Dr. Song Gao's lab in Sun Yat-sen University, a top lab for resolving crystal structure (Nature. 2017 Feb 16;542(7641):372-376, et.al.), we have been trying to resolve the co-crystal structure of MB-SHP2 complex. However, we have not been able to co-crystallize after trying many different methods. Without the structure information, we don't have binding details and are not able to validate them through point-mutating SHP2 gene.

We are still working on the co-crystal structure of MB-SHP2 complex. We hope that Reviewer could understand our situation.

Supporting Figure 9. MB directly binds SHP2 in vitro. SHP2-HIS protein was immobilized on a CM5 sensor chip by using standard amine-coupling at 25°C with running buffer HBS-P, a serial concentrations of compound MB was injected automatically. The binding signals were continuously recorded in response units (RU) and presented graphically as a function of time. The binding affinity of compound MB towards SHP2-HIS was assayed using the SPR-based Biacore T200 instrument. KD value was 56.45 μM for compound MB obtained by fitting the data sets to 1:1 Langmuir binding model using Biacore T200 Evaluation Software.

Minor concerns:

1. Fig. 3G, why *phos-CD28* weaker in the no PD-L1 conditions?

It could be that overexpressed PD-1 was eliciting baseline inhibitory signals in T cell. Indeed, we consistently found that PD-1 antibody, as well as MB, enhanced luciferase activity of just JP-luc seeded by its own in 96 well plate (Supporting Figure 10). This data suggested that overexpressed PD-1 underwent baseline activation, and that inhibition of this baseline inhibition leads to T cell activation. Along with this rationale, almost all of the PD-1 inhibitory activity in MB-treated T cells is cleared in comparison to baseline PD-1 activity in untreated T cells, which could explain stronger *phos-CD28* signal in OT-1 CTL co-cultured with Raji-PD-L1 in the presence of MB than that in OT-1 CTL co-cultured with parental Raji cells.

Supporting Figure 10: MB and PD-1 antibody enhanced luciferase activity of JP-luc cells. 5×10^4 JP-luc cell were seeded in 96-well plates. Cells were treated with 10 $\mu\text{g/ml}$ of nivolumab PD-1 antibody, or 1 μM MB respectively. Luciferase activity was assayed 6 after stimulation. JP-luc: Jurkat cell harboring NFAT-luciferase transgene and overexpressing human PD-1.

Data are representative of three independent experiments. Statistics were analyzed by unpaired t-test. Error bars denote s.e.m. *P < 0.05; **P < 0.01.

2. Fig. 1B, axis labels need to be fixed.

Thanks. The axis is now fixed.

Referee #2 (Comments on Novelty/Model System for Author):

The authors demonstrate the observed effect in several different model systems using murine and human cells in vitro, but also a murine tumor model in vivo. A variety of readout methods is utilized including FACS, classical Western Blot Analyses, and immunofluorescence. The clinical relevance is clear, as PD-1 PD-L1-blocking antibodies are currently the number one blockbuster in cancer therapy. The serious side effect profile, however obliges a search for alternatives. Fan and colleagues invented a very clever screening system for PD-1-related signaling with the intent to find small molecule inhibitors with similar therapeutic effects like the checkpoint-inhibiting antibodies Pembrolizumab and Nivolumab. With this system they identified Methylene Blue (MB), a substance used already in humans for other indications. They could convincingly show, that MB concentrations which could be reached easily within patients' blood were just as efficient as antibody-mediated inhibition of PD-1. They elucidated the molecular mechanism, showing that PD-1/SHP2 protein/protein-interaction is inhibited, resulting in interruption of the efferent signaling cascade. Their experiments show effectivity of inhibition both in vitro and in vivo. The clinical relevance of these findings is obvious. I have hence very little criticism:

Sometimes the authors use terms which are linguistically not correct, i.e. page 5: "...interactions (PPI) was previously ..." should be: "...interactions (PPI) were previously ...".

Thank so much. It has been corrected.

or on the next page: "... but shaded light on ..." must read: "... but shed light on ..."

Corrected.

On page 10 they use the word "trend". This makes the results weaker than they are, because trend is usually used when data do not reach statistical significance, although this is clearly the case here. I would suggest to use the word "pattern" instead.

Thanks so much! We have adopted "pattern".

There is one issue I must unfortunately address here: In figure 5A, the 3rd and 4th panel in the top row (24h PD-L1/Pem and 24h PD-L1/Niv are identical i.e. copy paste. I assume that was a mistake when generating the figure. This must be corrected.

We apologize for our mistake. We have corrected it.

Referee #3 (Comments on Novelty/Model System for Author):

The overall argument and science in this article seems outstanding and worthy of publication. I don't have the technical skills to review the scientific methods and when I contacted my colleague to get her help, she had sent the manuscript to me because she felt that she didn't have the skill set to review this manuscript. I would say that if the experimental work can be appropriately reviewed and looks okay, then I would publish it on all other counts. The person I think would be able to review this manuscript is Anusha Kalbasi anushakalbasi@mednet.ucla.edu I don't think it is fair to the authors for the paper to not be accepted because of my inability to review it. But I will say that two of us have struggled with it and that may indicate that it is not written at the right level. Likely too much detail is provided on how the authors got to the SHP2 result and the paper needs to be written to be more readable and less of a detailed account of all the experiments that were done. I am not sure though. I truly apologize, especially for the delay.

We appreciate these highly positive feedbacks from Reviewer.

2nd Editorial Decision

27th Mar 2020

Thank you for the submission of your revised manuscript to EMBO Molecular Medicine. We have now received the enclosed report from two referees who were asked to re-assess it. As you will see the referees are now supportive and I am pleased to inform you that we will be able to accept your manuscript pending the following amendments.

***** Reviewer's comments *****

Referee #1 (Remarks for Author):

The authors have fully addressed my concern. Great job!

Referee #2 (Comments on Novelty/Model System for Author):

The authors demonstrate the observed effect in several different model systems using murine and human cells in vitro, but also a murine tumor model in vivo. A variety of readout methods is utilized including FACS, classical Western Blot Analyses, and immunofluorescence. The clinical relevance is clear, as PD-1 PD-L1-blocking antibodies are currently the number one blockbuster in cancer therapy. The serious side effect profile, however obliges a search for alternatives. In the revised version in addition genetic evidence is provided for the specificity of the action of MB

Referee #2 (Remarks for Author):

The authors responded to all my comments in a very satisfying way; hence I consider the manuscript suitable for publication in its current form.

2nd Revision - authors' response

2nd Apr 2020

The authors performed the requested editorial changes.

Corresponding Author Name: Liang Chen, Penghui Zhou, Ke Ding

Manuscript Number: EMM-2019-11571-V2